

# No increase is detected and modeled for the seasonal cycle amplitude of $\delta^{13}$C of atmospheric carbon dioxide

Fortunat Joos[1,2], Sebastian Lienert[1,2], and Sönke Zaehle[3]

[1]Climate and Environmental Physics, University of Bern, Bern Switzerland
[2]Oeschger Centre for Climate Change Research, University of Bern, Bern, Switzerland
[3]Max Planck Institute for Biogeochemistry, P.O. Box 600164, Hans-Knöll-Str. 10, 07745 Jena, Germany

**Correspondence:** Fortunat Joos (fortunat.joos@unibe.ch)

**Abstract.** Measurements of the seasonal cycle of $\delta^{13}$C(CO$_2$) provide information on the global carbon cycle and the regulation of carbon and water fluxes by leaf stomatal openings on ecosystem and decadal scales. Land biosphere carbon exchange is the primary driver of $\delta^{13}$C(CO$_2$) seasonality in the Northern Hemisphere. We use isotope-enabled simulations of the Bern3D-LPX Earth System Model of Intermediate Complexity and fossil fuel emission estimates with a model of atmospheric transport to simulate local atmospheric $\delta^{13}$C(CO$_2$). Unlike the observed growth of the seasonal amplitude of CO$_2$ at northern sites, no significant temporal trend in the seasonal amplitude of $\delta^{13}$C(CO$_2$) is detected at most sites, consistent with the insignificant model trends. Comparing the preindustrial and modern periods, the modeled small amplitude changes at northern sites are linked to the near-equal increase of background atmospheric CO$_2$ and the seasonal signal of the net atmosphere-land $\delta^{13}$C flux in the northern extratropical region, with no long-term temporal changes in the isotopic fractionation by C3 plants. The good data-model agreement in the seasonal amplitude of $\delta^{13}$C(CO$_2$) and its decadal trend provides implicit support for the regulation of stomatal conductance by C3 plants towards intrinsic water use efficiency to grow proportionally to atmospheric CO$_2$ over recent decades. Disequilibrium fluxes contribute little to the seasonal amplitude of the net land isotope flux north of 40°N but contribute near-equally to the isotopic flux associated with growing season net carbon uptake in tropical and Southern Hemisphere ecosystems, pointing to the importance of monitoring $\delta^{13}$C(CO$_2$) over these ecosystems. We propose to apply seasonally-resolved $\delta^{13}$C(CO$_2$) observations as a novel constraint for land biosphere models and underlying processes for improved projections of the anthropogenic carbon sink.

## 1 Introduction

The seasonal variations in the carbon exchange fluxes between the atmosphere and the surface cause a seasonal cycle in atmospheric CO$_2$ (C$_a$) (Keeling et al., 1996; Graven et al., 2013; Masarie et al., 2014) and its stable isotopic signature ($\delta^{13}$C$_a$) (Keeling, 1960; Keeling et al., 1984, 1989, 2005; GLOBALVIEW-CO2C13, 2009). Observations of the atmospheric seasonal





cycles in background tropospheric air provide large-scale information on the carbon fluxes between the atmosphere, ocean, and land (Heimann et al., 1989, 1998) and constraints for models used to project atmospheric $CO_2$ and global warming.

The additional information of $\delta^{13}C$ data in comparison to carbon data stems from differences in fractionation for different carbon fluxes. Carbon isotopic fractionation describes the preferential transfer of light $^{12}C$ compared to heavier $^{13}C$. The degree of fractionation is different for the different physical, chemical, and biological processes (Mook, 1986) causing differences in the isotopic composition of carbon reservoirs and fluxes. The seasonal atmospheric $\delta^{13}C$ variations result from the combination of carbon and isotopic fluxes from fossil fuel burning, land use, and the exchange with the ocean, and land biosphere. Comparing results of carbon isotope-enabled models with observations of $\delta^{13}C_a$ is useful to assess whether the mix of carbon

and isotopic sink and source fluxes is represented consistently in comparison with the observations. $\delta^{13}C_a$ observations offer, therefore, a benchmark for evaluating and improving earth system models.

Fractionation is particularly large during the assimilation of $CO_2$ from the atmosphere by plants following the C3 photosynthesis pathway which are responsible for most of the global productivity (Still et al., 2003). Importantly, changes in isotopic fractionation by C3 plants are indicative of changes in stomatal conductance, regulating the leaf-internal $CO_2$ mixing ratio, and

thus photosynthesis (Farquhar, 1989; Saurer and Voelker, 2022; Cernusak and Ubierna, 2022). Photosynthesis, the associated water loss, and evaporative cooling are key characteristics of ecosystem function that are central to the cycles of carbon, nitrogen, water, and energy (Keenan et al., 2013; Knauer et al., 2017) and for the land sink of anthropogenic carbon. Acquisition of $CO_2$ for photosynthesis is accompanied by the loss of water through the stomatal pores that govern, by their conductance, the diffusion of these two gases between the leaf interior and the atmosphere. A key question is how ecosystems adjust their over-

all conductance and, thereby, co-regulate carbon uptake and plant growth, and water loss and evaporative cooling under rising atmospheric $CO_2$, growing nitrogen inputs to ecosystems, and increasing water vapor deficits under global warming. Many studies, relying on multi-decadal to century-scale tree-ring $\delta^{13}C$ records and free air $CO_2$ enrichment (FACE) experiments, suggest small changes in isotopic fractionation and intrinsic water use efficiency, the ratio of assimilation to conductance, to grow roughly proportionally with atmospheric $CO_2$ (Voelker et al., 2016; Saurer et al., 2014; Kauwe et al., 2013; Peñuelas

et al., 2011; Keller et al., 2017; Frank et al., 2015). In contrast, Battipaglia et al. (2013) and Keenan et al. (2013) suggest a scenario where conductance and the flows of carbon and water are downregulated under increasing $CO_2$. Conflicting results for 20th-century changes in fractionation and intrinsic water use efficiency are also found in global land biosphere models (Keller et al., 2017). Upscaling of results from site studies to large scales is challenging. It remains to be assessed whether a scenario with small long-term changes in fractionation of C3 plants is compatible with atmospheric $\delta^{13}C_a$ observations representing

carbon fluxes over large regions.

The observational records from globally distributed monitoring sites (Keeling et al., 1996; Graven et al., 2013; Masarie et al., 2014) demonstrate a significant growth trend in the seasonal cycle amplitude ($SA$) of $C_a$ (Keeling et al., 1996; Graven et al., 2013). The observed seasonal cycle and amplitude growth of $C_a$ are widely used to evaluate carbon cycle models and system understanding by transporting fluxes from terrestrial, oceanic, and fossil sources with a model of atmospheric transport

to obtain local $C_a$ anomalies (Heimann et al., 1998; Dargaville et al., 2002; Scholze et al., 2008; Peng et al., 2015; Lienert and Joos, 2018). Studies address the role of different climatic drivers and terrestrial carbon cycle processes such as drought,



land use, warming, productivity, and soil respiration (Heimann et al., 1989, 1998; Graven et al., 2013; Forkel et al., 2016; Ito et al., 2016; Bastos et al., 2019; Wang et al., 2020) and surface-to-atmosphere C fluxes (e.g. Peylin et al. (2013)). $SA(C_a)$ and their temporal trends at different monitoring sites are used for constraining an ensemble of land biosphere model simulations

(Lienert and Joos, 2018).

Comparable studies, analyzing the temporal trends in $SA(\delta^{13}C_a)$ and the seasonal cycle of $\delta^{13}C_a$, are, to our knowledge, lacking. While seasonally-resolved atmospheric $\delta^{13}C_a$ measurements are available (GLOBALVIEW-CO2C13, 2009; Keeling et al., 2001), these seasonally-resolved records are yet to be fully utilized in the context of processed-based carbon cycle models. Heimann et al. (1989) simulated the spatiotemporal distribution of $\delta^{13}C_a$ and $C_a$ with an atmospheric transport model

using estimates of net primary productivity and heterotrophic respiration based on satellite data and surface temperature and prescribed surface ocean $CO_2$, demonstrating the dominant role of land biosphere fluxes for northern hemisphere seasonality and finding relevant signals from the ocean and land in the southern hemisphere. Observations of $\delta^{13}C_a$ seasonal cycles have been used to investigate isotopic fractionation (Ballantyne et al., 2010) and trends in the phenology of northern terrestrial ecosystems (Gonsamo et al., 2017), but to our knowledge have not been used as a benchmark for model performance in

combination with an atmospheric transport model and for analyzing trends in $SA(\delta^{13}C_a)$ globally.

This study addresses the following main questions:

1. Is the seasonal cycle of $\delta^{13}C_a$ observed at a network of globally distributed sites well represented in model simulations? How large are the contributions of ocean, land, and fossil fuel fluxes to $\delta^{13}C_a$ seasonality?

2. What are the temporal trends in $SA(\delta^{13}C_a)$ in the observational records and are the modeled trends in $SA(\delta^{13}C_a)$

consistent with the observed trends?

3. What are the different drivers of $SA(\delta^{13}C_a)$ versus $SA(C_a)$ and of their temporal trends? Is a model scenario with intrinsic water use efficiency growing proportional with atmospheric $CO_2$ consistent with $\delta^{13}C_a$ data?

We simulate atmospheric $\delta^{13}C_a$ and $C_a$ at 19 sites using the matrix representation of an atmospheric transport model and net atmosphere-to-surface fluxes of $CO_2$ and $\delta^{13}(CO_2)$ from an Earth System Model of Intermediate Complexity (EMIC)

alongside gridded fossil fuel emission estimates and changes in land use and the distribution of C3 and C4 crops. We compare model results to observations and analyze trends in $SA(\delta^{13}C_a)$ using the records of the Scripps $CO_2$ program (Keeling et al., 2001) and the Cooperative Global Atmospheric Data Integration Project (2013) product. We demonstrate for the first time that the observations at the globally distributed sites show no significant trends in the seasonal cycle amplitude of $\delta^{13}C_a$, consistent with our model chain, but surprising in view of the large trend in the seasonal amplitude of $CO_2$. We discuss the implications

of our results for changes in the fractionation by C3 plants, their stomatal controls, and associated carbon and water fluxes. We develop a theoretical framework to explain the trends in $SA(\delta^{13}C_a)$ and decompose net carbon and isotope land biosphere fluxes into underlying component fluxes and changes in carbon fluxes and fractionation. The framework could serve future studies, e.g., studies applying an ensemble of different models for multi-model evaluation and more robust conclusions in comparison to using a single model chain.



## 2 Methods

### 2.1 Bern3D-LPX

Spatially-resolved surface-to-atmosphere $CO_2$ and $^{13}CO_2$ fluxes are simulated with the EMIC Bern3D-LPX. Here, the ocean-atmosphere model Bern3D (Jeltsch-Thömmes and Joos, 2020; Battaglia and Joos, 2018; Ritz et al., 2011) is coupled to the Dynamic Global Vegetation Model (DGVM) LPX-Bern v1.4 (Lienert and Joos, 2018). The Bern3D model features a 41 x 40 horizontal ocean resolution (about $9^o$x$4.5^o$) with 32 depth layers, coupled to a single-layer energy-moisture balance atmosphere (Ritz et al., 2011). In Bern3D, carbon and its isotopes are implemented as tracers with fractionation for air-sea and sea-air gas exchange, aquatic chemistry, and the production of organic material and $CaCO_3$ as a function of surface ocean temperature, aqueous $CO_2$, and the speciation of DIC as described by Jeltsch-Thömmes and Joos (2023). LPX-Bern simulates the coupled cycling of carbon, nitrogen, and water (Xu-Ri and Prentice, 2008; Wania et al., 2009a, b; Stocker et al., 2014) and vegetation dynamics using plant functional types (Sitch et al., 2003). It is here run on a $3.75°$x$2.5°$ resolution. Grid cells are subdivided into different land use classes (mineral soil, wetlands, crop, pasture, urban). Carbon isotopes were added (Scholze et al., 2003) using a photosynthetic fractionation scheme (Lloyd and Farquhar, 1994) and without further isotopic fractionation during the transfer through vegetation, litter, soil, and product pools. The signature of respired carbon reflects the signature of carbon assimilated at previous times; the lag times between assimilation and respiration are dictated by the turnover time scales of the various pools, depending on temperature and soil moisture. Land carbon and isotope fluxes respond to altered climate, which influences, for example, photosynthesis through temperature and water limitation, fire frequency, and autotrophic and heterotrophic respiration rates, to increasing $CO_2$, which stimulates photosynthesis and affects water use efficiency ("$CO_2$ fertilization"), and to land use (Strassmann et al., 2008), which causes, for example, transfer of tree carbon to the atmosphere, litter, and product pools after deforestation and shifts from natural vegetation to C3 and C4 crops and pasture, and to altered nitrogen deposition and nitrogen fertilizer addition on managed land alleviating nitrogen limitation.

Bern3D and LPX-Bern were spun up individually, followed by a 500-year coupled spinup to pre-industrial equilibrium (1700 CE; 276.3 ppm, -6.27 ‰). A transient simulation, $E_{standard}$, from 1700 to 2020 is driven by annual fossil carbon emissions (including the contribution from cement production) (Friedlingstein et al., 2020), net land use area changes (Hurtt et al., 2020), and non-$CO_2$ radiative forcing. $\delta^{13}C$ of the fossil fuel emissions follows Andres et al. (2017) for 1751-2014 and set to the value for 1751 before. For 2014-2020, signatures of major source categories (coal, oil, gas, cement) are assumed constant and combined with the emission sources from Friedlingstein et al. (2020), following the approach of Andres et al. (2000). Here, we explicitly distinguish land use classes for C3 and C4 crops and prescribe their extent, and net land use area changes, based on LUH2 (Hurtt et al., 2020). Nitrogen deposition and nitrogen fertilization are taken from the NMIP project (Tian et al., 2018). Nitrogen (N) is a limiting nutrient in LPX and plant growth is downregulated under N-stress, which tends to reduce plant growth and plant growth responses to rising $CO_2$ compared to a model with absent N cycling. The NCEP/NCAR monthly wind stress climatology (Kalnay et al., 1996) is prescribed to the ocean. Monthly climate fields from CRU-TS4.05 (Harris et al., 2020) are used for the land model. For 1700-1900 and the spinup, the climate of 1901-1931 is recycled. A control simulation ($E_{control}$) without anthropogenic $CO_2$ emissions, and absent radiative forcing from non-$CO_2$ agents (e.g., from $CH_4$, $N_2O$,



ozone), land use, nitrogen deposition, and nitrogen fertilization at 1700 level, as well as recycling 1901-1931 land climate

provides a baseline. Atmospheric $CO_2$ and $\delta^{13}C$ evolve freely in all simulations presented and remain at their preindustrial

values in $E_{control}$.

## 2.2  Atmospheric Transport Model TM3 and the seasonal cycles of $CO_2$ and $\delta^{13}C_a$

We employ the transport matrices of the TM3 atmospheric transport model (Heimann and Körner, 2003; Kaminski et al., 1998;

Schürmann et al., 2016) to translate surface-atmosphere fluxes from Bern3D-LPX and fossil emissions into $CO_2$ ($C_a$) and

$\delta^{13}C(CO_2)$ ($\delta^{13}C_a$) anomalies at 19 measurement sites across the globe. Before transport, the fluxes are remapped to the TM3

72x48 grid ($5^o \times 3.75^o$). Here, the matrices span from 1982 to 2012 and are only available if there is also a $CO_2$ measurement

available at the corresponding site. Each matrix represents the sensitivity of the local atmospheric concentration for a given

month to the local surface fluxes of the previous period, spanning up to 48 months. The transport model is initialized with

equal $C_a$ and $\delta^{13}C_a$ at all sites.

For $^{13}C$, the signature-weighted net atmosphere-to-surface flux is:

$$\delta^{13}f_{as,net}(\mathbf{x},t) = f_{as,net}(\mathbf{x},t) \cdot \delta^{13}C_{as,net}(\mathbf{x},t). \tag{1}$$

$\delta^{13}f_{as,net}$ is in units of mol permil m$^{-2}$ yr$^{-1}$. $\mathbf{x}$ indicates location and $t$ time at the monthly and spatial ($5^o \times 3.75^o$) resolution

of TM3. The net carbon fluxes ($f_{as,net}$; mol m$^{-2}$ yr$^{-1}$), their signatures ($\delta^{13}C_{as,net}$), and, therefore, $\delta^{13}f_{as,net}$, are readily

available for fossil emissions, including cement production (Andres et al., 2009b, a). Bern3D-LPX simulates two-way exchange

of $CO_2$ and $^{13}CO_2$ from and to the ocean and land surface. Net transfer rates are determined by the difference of these gross

fluxes to yield atmosphere-to-surface net fluxes $f_{as,net}$ and $\delta^{13}f_{as,net}$ of Bern3D-LPX.

The matrices are applied with $f_{as,net}$ to compute anomalies in $C_a$ and with $\delta^{13}f_{as,net}$ to compute anomalies in $^{13}CO_2$.

We get $\delta^{13}C_a$ from $^{13}CO_2/C_a$. This method of transporting signature-weighted net fluxes was chosen instead of separately

transporting $^{13}CO_2$ and $^{12}CO_2$. Both approaches were tested and showed very similar results, except for numerical issues in

months having very small local $^{12}CO_2$ anomalies for the second approach.

Ocean, land, and fossil fluxes from the standard simulation are transported separately to quantify the contributions of these

individual components to the seasonal variations in $C_a$ and $\delta^{13}C_a$. For $E_{control}$, fossil fuel fluxes are not transported, consistent

with the model setup.

## 2.3  Site data

Observations from 19 monitoring sites are used for comparison with simulated $C_a$ and $\delta^{13}C_a$ and to determine observation-

based trends in their $SA$. The Cooperative Global Atmospheric Data Integration Project (2013) product is used for $C_a$. For

$\delta^{13}C_a$, the records of the Scripps $CO_2$ program (Keeling et al., 2001) for Alert, Mauna Loa, and the South Pole from monthly-

averaged flask data are used. These records span a longer period than the available transport matrices. For the remaining 16

sites, the shorter (1994 to 2009) records of GLOBALVIEW-CO2C13 (2009) are used. In the main manuscript, we focus on 3 out





of the 19 available transport sites: Alert (82.5°N, Canada), Mauna Loa (19.5°N, Hawaii), and South Pole (90°S, Antarctica). Results for the other sites are shown in the supplementary and Table 1.

Additional $\delta^{13}C_a$ monthly flask data from the Scripps $CO_2$ program (Keeling et al., 2001) are used for analyzing temporal trends in the $SA(\delta^{13}C_a)$. We focus on eight sites with more than twenty years of data: Alert (ALT, 82°N), Point Barrow (PTB, 71°N), La Jolla (LJO, 33°N), Mauna Loa Observatory (MLO, 20°N), Cape Kumukahi (KUM, 20°N),Christmas Island (CHR, 160 2°N), Samoa (SAM, 14°S), and South Pole (SPO, 90°S). The records from Kermadec Island (KER, 29°S) and Baring Head, New Zealand (NZD, 41°S) extend over more than 20 years, but have often missing monthly values.

## 3 The influence of carbon and isotope fluxes on the seasonal cycles of $CO_2$ and $\delta^{13}CO_2$: a conceptual framework

We develop a simplified conceptual framework to qualitatively explore the influence of carbon and isotope fluxes on the seasonal cycles of $C_a$ and $\delta^{13}C_a$. For illustration, the atmosphere is considered to be well mixed in this section; the atmospheric 165 transport operator is linear and the findings may qualitatively also apply to spatially-resolved fluxes. The budgets for the atmospheric inventories of carbon and $^{13}C$ are approximated (Tans et al., 1993):

$$\frac{d}{dt}N_a = -F_{as,net} \tag{2}$$

$$\frac{d}{dt}(N_a \cdot \delta^{13}C_a) = \left(\frac{d}{dt}N_a\right)\cdot\delta^{13}C_a + N_a\cdot\left(\frac{d}{dt}\delta^{13}C_a\right) = -F_{as,net}\cdot\overline{\delta^{13}C_{as,net}} \tag{3}$$

$N_a$ and $N_a\cdot\delta^{13}C_a$ are the atmospheric inventories of carbon and (approximately) of $^{13}C$ (in mol permil). $F_{as,net}$ and $F_{as,net}\cdot\overline{\delta^{13}C_{as,net}}$ are the globally integrated net atmosphere-to-surface carbon and $^{13}C$ flux; $\overline{\delta^{13}C_{as,net}}$ is the signature of the global net carbon flux. We set $N_a = c\cdot C_a$, where $c$ is a unit conversion factor. Solving Eqs. 2 and 3 for the change in $C_a$ and $\delta^{13}C_a$ yields:

$$\frac{d}{dt}C_a = \frac{-1}{c}\cdot F_{as,net} \tag{4}$$


$$\frac{d}{dt}\delta^{13}C_a = \frac{-1}{c\cdot C_a}\cdot\delta^{13}F^*_{as,net}, \tag{5}$$

with $\delta^{13}F^*_{as,net}$ being the global integral of

$$\delta^{13}f^*_{as,net} = f_{as,net}\cdot\left(\delta^{13}C_{as,net} - \delta^{13}C_a\right) \tag{6}$$

The superscript * indicates that the $^{13}C$ fluxes (e.g., in units of mol permil yr$^{-1}$ m$^{-2}$ for $\delta^{13}f^*_{as,net}$) are referenced to the 180 atmospheric signature. Eq. 6 corresponds to Eq. 1 for the net atmosphere-to-surface isotopic flux but is now referenced to the atmospheric signature instead the signature of 0 permil of the Vienna Pee Dee Belemnite standard as in Eq. 1. In this way, a positive (negative) flux causes a positive (negative) change in $\delta^{13}C_a$.





Eqs. 4 and 5 are readily integrated over the growing season from the intraannual maximum to the minimum (subscripts max, min) in $C_a$ and the corresponding beginning, $t_{beg}$, and end time, $t_{end}$, of the growing season to get the seasonal cycle amplitude

($SA$) for the two tracers and (cumulative) net fluxes (see Appendix A for calculation of $SA$ for a flux):

$$\underbrace{C_{a,max} - C_{a,min}}_{SA(C_a)} = \frac{1}{c} \underbrace{\int_{t_{beg}}^{t_{end}} dt\, F_{as,net}(t)}_{SA(F_{as,net})} \tag{7}$$

$$\underbrace{\delta^{13}C_{a,max} - \delta^{13}C_{a,min}}_{SA(\delta^{13}C_a)} = \frac{-1}{c \cdot C_a} \underbrace{\int_{t_{beg}}^{t_{end}} dt\, \delta^{13}F_{as,net}^*(t)}_{SA(\delta^{13}F_{as,net}^*)} \tag{8}$$

Eqs. 5 and 8 provide important insight. First, changes in $\delta^{13}C_a$ and its seasonal cycle are driven by $\delta^{13}F_{as,net}^*$; seasonal

changes in $C_a$, the denominator in Eq. 5, are small compared to $C_a$ and $C_a$ considered constant within a given year (the error associated with this approximation is less than 3%). Second, the background $CO_2$ mixing ratio, $C_a$, modulates the magnitude of the $\delta^{13}C_a$ seasonal cycle. $SA(\delta^{13}C_a)$ would be larger under low preindustrial $CO_2$ than under modern $CO_2$ for equal seasonal variations in $\delta^{13}F_{as,net}^*$. Correspondingly, $SA(\delta^{13}C_a)$ does not change over time as long as relative changes in $SA(\delta^{13}F_{as,net}^*)$ and in $CO_2$ are equal. Equations 7 and 8 were derived for a globally well mixed atmosphere and global fluxes, but analogously

also apply for the tracer seasonality at individual sites with the integral on the right-hand side of Eqs. 7 and 8 representing the integral of (transport-weighted) fluxes over the region influencing tracer seasonality at the site. We recall that the above equations and conclusions were derived by assuming a well-mixed atmosphere, while in reality spatial flux patterns and transport and their changes influence seasonal cycles at individual atmospheric sites. Further, the start and end of the growing season are assumed to coincide with the switch in the sign of the isotopic flux; this is the case in our model for zonally integrated fluxes.

These seasonal fluxes will be presented in section 3.3. Equations 2 to 8 are for illustrating the influence of carbon and carbon isotope fluxes on the seasonal cycles of $CO_2$ and $\delta^{13}CO_2$; they were not used for calculating numerical results.

The notation and sign convention introduced above are applied in this manuscript. In brief, $f_{i,j}$ defines a one-way flux from the source reservoir $i$ to the receiving reservoir $j$ and is positive. The isotopic signature of this flux is $\delta^{13}C_{i,j}$. The net flux from reservoir $i$ to reservoir $j$ is $f_{i,j,net}$ and is the difference between the corresponding one-way fluxes, e.g., $f_{i,j,net}=f_{i,j}-f_{j,i}$.

$f_{i,j,net}$ is positive if the net flux results in the transfer of mass from $i$ to $j$.

## 4 Results

### 4.1 Seasonal cycles of atmosphere-surface fluxes, $C_a$, and $\delta^{13}C_a$

The model simulates large seasonal variations in the net land biosphere-atmosphere exchange of $CO_2$ and $^{13}CO_2$, whereas seasonal variations in ocean-atmosphere fluxes are much smaller (Fig. 1). The seasonality of the Bern3D-LPX carbon fluxes




**Table 1.** The seasonal cycle amplitude of $C_a$ and $\delta^{13}C_a$ from the standard simulation (Std; $E_{standard}$) and observations (Obs) for 19 monitoring sites and the period 1982-2012. The increase over the industrial period is estimated from the difference between the standard simulation and the preindustrial control ($100*(E_{standard} - E_{control})/E_{control}$).

| Site | Seasonal Cycle Amplitude of $C_a$ | | | | Seasonal Cycle Amplitude of $\delta^{13}C_a$ | | | |
| | Standard | Observed | Std−Obs | Increase | Standard | Observed | Std−Obs | Increase |
| | [ppm] | [ppm] | [ppm] | [%] | [permil] | [permil] | [permil] | [%] |
| --- | --- | --- | --- | --- | --- | --- | --- | --- |
| Alert, Nunavut, Canada (82°N) | 17.26 ± 0.84 | 14.82 ± 0.75 | 2.44 | 41 | 0.719 ± 0.035 | 0.750 ± 0.042 | -0.030 | 8 |
| Barrow, Alaska, US (71°N) | 20.10 ± 1.06 | 15.79 ± 0.75 | 4.31 | 37 | 0.899 ± 0.041 | 0.822 ± 0.030 | 0.077 | 6 |
| Ocean Station M, Norw. (66°N) | 17.73 ± 1.14 | 14.77 ± 0.93 | 2.96 | 36 | 0.763 ± 0.051 | 0.752 ± 0.028 | 0.010 | 4 |
| Cold Bay, Alaska, US (55°N) | 14.83 ± 0.80 | 15.91 ± 1.00 | -1.08 | 39 | 0.657 ± 0.032 | 0.847 ± 0.029 | -0.190 | 4 |
| Shemya Island, Alaska, US (53°N) | 14.63 ± 0.88 | 17.03 ± 0.98 | -2.40 | 37 | 0.655 ± 0.038 | 0.926 ± 0.029 | -0.271 | 3 |
| Mace Head, Ireland (53°N) | 14.65 ± 1.15 | 13.64 ± 1.74 | 1.01 | 42 | 0.618 ± 0.046 | 0.726 ± 0.039 | -0.107 | 6 |
| Terceira Island, Portugal (39°N) | 11.70 ± 0.86 | 9.39 ± 1.18 | 2.31 | 36 | 0.497 ± 0.028 | 0.517 ± 0.036 | -0.020 | -1 |
| Key Biscayne, Florida, US (26°N) | 8.97 ± 0.78 | 8.17 ± 2.31 | 0.79 | 33 | 0.350 ± 0.031 | 0.405 ± 0.040 | -0.055 | -6 |
| Mauna Loa, Hawaii, US (20°N) | 8.33 ± 0.30 | 6.51 ± 0.24 | 1.82 | 38 | 0.328 ± 0.013 | 0.339 ± 0.028 | -0.010 | 4 |
| Cape Kumukahi, Hawaii (20°N) | 9.07 ± 0.39 | 7.96 ± 0.53 | 1.11 | 41 | 0.358 ± 0.017 | 0.423 ± 0.028 | -0.065 | 5 |
| Mariana Islands, Guam (13°N) | 6.36 ± 0.41 | 6.11 ± 0.72 | 0.26 | 39 | 0.254 ± 0.019 | 0.312 ± 0.022 | -0.058 | 3 |
| Ragged Point, Barbados (13°N) | 8.00 ± 0.39 | 7.10 ± 0.45 | 0.89 | 38 | 0.306 ± 0.017 | 0.332 ± 0.027 | -0.026 | 9 |
| Christmas Island, Kiribati (2°N) | 4.68 ± 0.35 | 2.95 ± 0.38 | 1.74 | 77 | 0.161 ± 0.016 | 0.129 ± 0.018 | 0.033 | 52 |
| Ascension Island, UK (8°S) | 3.17 ± 0.41 | 1.81 ± 0.44 | 1.37 | 8 | 0.134 ± 0.012 | 0.073 ± 0.016 | 0.061 | -28 |
| Mahe Island, Seychelles (5°S) | 3.52 ± 0.41 | 2.57 ± 0.53 | 0.95 | 175 | 0.145 ± 0.015 | 0.113 ± 0.021 | 0.032 | 63 |
| Tutuila, American Samoa (14°S) | 1.29 ± 0.28 | 0.89 ± 0.30 | 0.40 | 51 | 0.044 ± 0.007 | 0.020 ± 0.009 | 0.024 | -8 |
| Palmer Station, Antarctica (65°S) | 2.60 ± 0.25 | 1.41 ± 0.19 | 1.20 | 21 | 0.107 ± 0.008 | 0.037 ± 0.008 | 0.070 | -17 |
| Halley Station, Antarctica (76°S) | 2.57 ± 0.15 | 1.09 ± 0.16 | 1.48 | 32 | 0.103 ± 0.005 | 0.027 ± 0.007 | 0.076 | -5 |
| South Pole, Antarctica (90°S) | 2.06 ± 0.16 | 1.09 ± 0.11 | 0.97 | 20 | 0.094 ± 0.004 | 0.033 ± 0.015 | 0.061 | -4 |





**Figure 1.** Net seasonal atmosphere-to-surface fluxes. Fluxes are for (a,c,e) carbon and (b,d,f) the $\delta^{13}$C-weighted carbon flux, $\delta^{13} f^*_{as,net}$ (see section 3) from the standard simulation ($E_{standard}$) and averaged over 1982-2012 for (a,b) June, July, August (JJA), (c,d) December, January, February (DJF), and (e,f) JJA minus DJF. Note non-linear color bars with blue colors in panels a to d indicating a lowering in atmospheric $CO_2$ and $\delta^{13}$C, respectively.





is broadly consistent with estimates of regional air-land carbon flux seasonality from an atmospheric inversion (Gurney et al., 2004) and air-sea flux seasonality from surface ocean $pCO_2$ observations (Fay et al., 2021), except in the Southern Ocean. The land biosphere model shows the expected uptake of isotopically depleted carbon, resulting in positive $f_{as,net}$ and negative $\delta^{13}f^*_{as,net}$, during the summer and vice versa in winter. On a separate note, the nominal signature of the net carbon flux, $\delta^{13}C_{as,net}$, may be diagnosed by dividing $\delta^{13}f_{as,net}$ by $f_{as,net}$. $\delta^{13}C_{as,net}$ can become very large for small $f_{as,net}$, is hard to interpret, and not used in our approach.

The ocean model shows a negative $\delta^{13}f^*_{as,net}$ in low and mid-latitudes, small modern fluxes in the northern subpolar gyres, and a positive flux in the Southern Ocean in both seasons (Fig. 1). These modern Bern3D fluxes are driven by the atmosphere-ocean isotopic disequilibrium, here defined as the isotopic signature of the atmosphere-to-surface carbon flux minus the signature of the surface-to-atmosphere flux ($\delta_{dis,as}$; Eq. A2), with a negative $\delta_{dis,as}$ in low and mid-latitudes, a small modern disequilibrium in northern high latitudes, and a positive $\delta_{dis,as}$ south of $50^oS$, consistent with observations (Menviel et al., 2015; Quay et al., 2017; Becker et al., 2018).

The preindustrial air-sea isotopic disequilibrium ($\delta_{dis,as}$; see Appendix A) and $\delta^{13}f^*_{as,net}$ are negative in low- and mid-latitude and positive in high-latitude ocean regions (not shown), mainly driven by the temperature dependency of isotopic fractionation during air-sea exchange and the cycling of marine biological matter (see Fig. 1 of Menviel et al. (2015) for a comparison of Bern3D and LOVECLIM results for the atmosphere-ocean disequilibrium). Fossil fuel emissions cause a negative flux perturbation worldwide, shifting the net isotopic fluxes to more negative values over the industrial period.

Figure 2 compares the mean seasonal cycles of $C_a$ and $\delta^{13}C_a$ from the standard run with measurements from 1982 (Alert: 1985) to 2012 at three sites, and with factorial simulations, where the fluxes of land (green dashed line), the ocean (blue dashed line), and fossil fuel emissions (brown dashed lines) were considered individually (See Table 1 and Supplementary Information, Fig. S1 and S2 for additional sites). For the Northern Hemisphere (NH) sites of Alert (top panels) and Mauna Loa (middle panels), the seasonal variations are dominated by the terrestrial biosphere fluxes, with minor contributions from ocean fluxes and fossil fuel emissions. Note that in contrast to the seasonality of $C_a$, the seasonality of $\delta^{13}C_a$ does not linearly decompose into the contributions of land, ocean, and fossil fuel emissions. Both the timing and amplitude of the observed seasonal cycle of $C_a$ and $\delta^{13}C_a$ are captured reasonably well by the standard simulation. The simulated $SA(C_a)$ and its interannual variability (IAV) are overestimated compared to observations at Alert (17.3±0.84 ppm vs 14.8±0.75) and Mauna Loa (8.3±0.30 ppm vs 6.5±0.24 ppm). $SA(\delta^{13}C_a)$ matches the observations (ALT: 0.72±0.035 ‰ vs 0.75±0.042 ‰; MLO: 0.34±0.013 ‰ vs 0.33±0.028). Good model-data agreement in the phasing of the seasonal cycle of $C_a$ relative to $\delta^{13}C_a$ is demonstrated for Alert in panel (c), where monthly anomalies in $\delta^{13}C_a$ are plotted versus anomalies in $C_a$. Both observation and model show hysteresis throughout the year, with the loop rotating clockwise. At Mauna Loa, the rotation direction of the hysteresis loop is clockwise in the simulation and anticlockwise in the observation (panel (f)), but the observed hysteresis is small with offsets of less than 0.03 ‰. The hysteresis arises as the ratio between the rate of change in $\delta^{13}C_a$ versus the rate of change in $C_a$ varies over the year (Keeling et al., 1989; Heimann et al., 1989). This non-linearity in the atmospheric tracer relationship originates from seasonally varying transport in combination with spatially and temporally varying relationships of atmosphere-surface $\delta^{13}C$ to $CO_2$ flux. For example, the isotopic signature of the growing season net atmosphere-to-land carbon flux $\delta^{13}C_{al,net}$ is





**Figure 2.** The simulated (red) seasonal cycle of atmospheric $CO_2$ (left (a,d,g)) and its signature $\delta^{13}C_a$ (middle (b,e,h)), compared to observations (black dots). In the rightmost panels (c,f,i) the seasonal anomalies ($\Delta$) of $CO_2$ are plotted against those of $\delta^{13}C_a$, with lines connecting the monthly values (dots) fading from January to December. Results are for Alert, northern Canada (a,b,c), Mauna Loa, Hawaii (d,e,f), and the South Pole (g,h,i). Simulated values are from transporting in TM3 net fluxes of the Bern3D-LPX standard simulation from all (red, $E_{standard}$), terrestrial (green, dashed), oceanic (blue, dashed), and fossil sources (brown, dashed). The observational and model anomalies are computed from monthly values between 1982 and 2012 if both the measurements and transport matrices are available. Error bars and shading correspond to the standard deviation from the interannual variability of monthly values.



-13.4 ‰ for the northern high-latitude region (>40°N), but only -10.7 ‰ for the region 10°N-40°N and the signal observed at any measurement site results from varying contributions from these and other latitudinal bounds given intraannually varying winds and hence transport.

     Results for the South Pole are different than for the NH sites (Fig. 2 (g,h,j)). Neither the timing nor the amplitude of $C_a$ (2.1±0.16 ppm simulated vs 1.1±0.11 ppm observed) and $\delta^{13}C_a$ (0.094±0.004 ‰ vs 0.033±0.015 ‰) agree with observa-

tions. $SA(C_a)$ and $SA(\delta^{13}C_a)$ at the South Pole are observed to be 14 and 23 times smaller than at Alert, respectively. The absolute data-model mismatches are therefore not as drastic as the relative mismatches. The disagreement between simulation and observational estimates is also apparent when considering the scatter plot in panel (i). The model shows a complex hysteresis relationship, whereas the observation displays a clockwise loop.

     The remote Antarctic sites (South Pole, Palmer, Halley) show an expected relatively larger dependence on the ocean, but the

terrestrial contribution still dominates in the model (Figs. 2, S1, S2). The $C_a$ seasonal cycle resulting from atmosphere-ocean flux is shifted by up to six months compared to observations at the South Pole and the other two Antarctic sites (Palmer, Halley; blue lines versus black dots in Fig. S1), pointing to biases in the Bern3D ocean flux. The Bern3D ocean fluxes and their seasonality are broadly comparable to observational estimates over large parts of the ocean (Landschützer et al., 2014; Takahashi et al., 2009; Fay et al., 2021), with ocean $CO_2$ outgassing in the tropics and uptake in the mid-latitudes during winter

(Fig. 1). However, observation-based analyses indicate stronger ocean $CO_2$ uptake in summer than in winter in the Southern Ocean (Jin et al., 2024; Long et al., 2021; Fay et al., 2021) and the northern subpolar gyres, in contrast to results from Bern3D (Fig. 1) and more complex ocean models (Hauck and Völker, 2015) and several Earth System Models from CMIP5 (Majkut et al., 2014) and CMIP6 (Joos et al., 2023). The simulated amplitude and phasing of the $\delta^{13}C_a$ seasonal cycle resulting from the ocean are broadly in line with observations at the Antarctic sites (Fig. S2). The air-sea isotopic disequilibrium is large in the

Southern Ocean and the two-way, air-sea and sea-air, exchange fluxes yield a substantial net isotopic flux, even under low net carbon flux. Temperature-dependent fractionation is higher in winter than summer and the air-sea gas exchange piston velocity, and, in turn, the isotope fluxes are larger under high winds in winter than in summer in the model Southern Ocean, consistent with the observed seasonal phasing of $\delta^{13}C_a$ at the Antarctic sites. Errors in modeled Southern Ocean fluxes are expected to have a minor impact on simulated $SA(C_a)$ and $SA(\delta^{13}C_a)$ at NH sites, where the influence of land fluxes dominates by far

(Fig. 2, S1, S2).

     Considering all extratropical Northern Hemisphere sites, model-data mismatches are less than 30% for $SA(C_a)$ and $SA(\delta^{13}C_a)$ and their root mean square errors (RMSE) are 2.6 ppm and 0.14 permil, respectively. For the tropical and SH sites, large relative data-model deviations of up to 140% for $SA(C_a)$ and up to 290% for $SA(\delta^{13}C_a)$ are evident, although absolute deviations are less than 1.8 ppm and 0.18 permil and the corresponding RMSEs are 1.2 ppm and 0.05 permil (Table 1).

Interannual variability in simulated $SA(\delta^{13}C_a)$ compares reasonably well with observations at sites in the NH subtropics and extratropics (average of the 1-$\sigma$ standard devation of 12 sites: 0.031 ‰ in $E_{standard}$ versus 0.031 ‰ in observations) and in the tropics and SH (0.009 vs. 0.013 ‰) (Table 1). Similar agreement between simulated and observation-derived IAV holds for $SA(C_a)$ (NH extratropics: 0.75 vs 0.96 ppm; tropics and SH: 0.29 vs 0.30). This further suggests that the variability in the seasonal amplitude of the underlying carbon and isotope fluxes is reasonably represented by LPX-Bern. The correct simulation



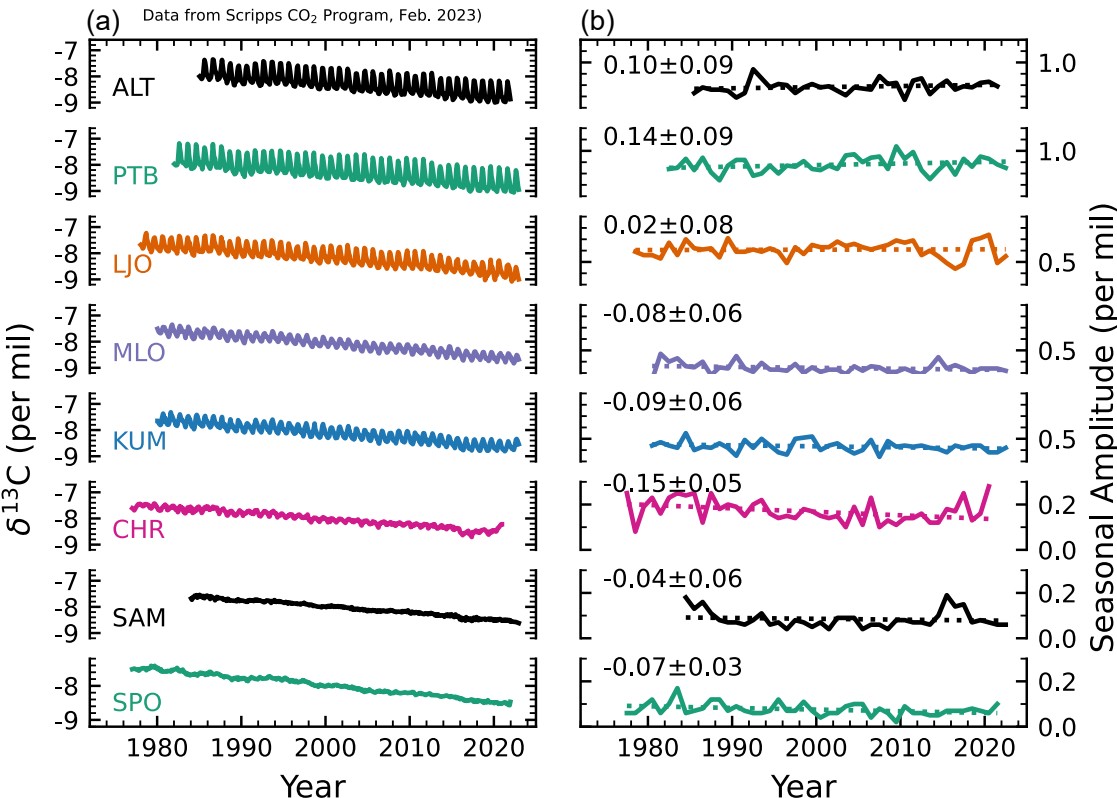

**Figure 3.** Temporal evolution of $\delta^{13}C_a$ (left) and its seasonal amplitude (right) from data of the Scripps network (Keeling et al., 2001). Gap-filled data provided by Scripps are used for the eight sites. The slope and its standard error from a linear regression through the seasonal amplitude data (dotted) are given in permil/century. Trends are not different from zero based on a two-sided t-test and a significance level of 5%, except at Christmas Island (CHR) and the South Pole (SPO).

of variability can be challenging and van der Velde et al. (2013) report too low interannual variability in the annually-integrated isotopic disequilibrium flux for their model.

## 4.2    Temporal trends in the seasonal cycle amplitude of $\delta^{13}C_a$ and $C_a$

Temporal trends in $SA(\delta^{13}C_a)$ from the Scripps gap-filled data are not statistically different from zero, except at the tropical site Christmas Island and the South Pole (Fig. 3). Averaging the trends across all 8 sites yields -0.0038±0.026 permil/century

(mean± 1 sdv of mean) and averaging the trends for the extratropical sites ALT, PTB, and LJO yields +0.09±0.06 permil/-century, with both averaged trends not statistically different from zero. The trend for the NH extratropical sites translates into a change in $SA(\delta^{13}C_a)$ of around 5±3% over the 40-year observational period. Detection of trends in $SA(\delta^{13}C_a)$ may be

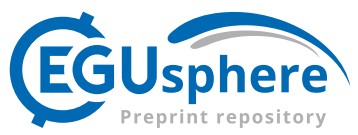

**Table 2.** Temporal trends in the seasonal cycle amplitude of $C_a$ and $\delta^{13}C_a$ from the standard simulation ($E_{standard}$) and observations for 19 monitoring sites from 1982–2012. Observational data of $CO_2$ are from the GLOBALVIEW-CO2 product and fitted for the period 1982–2012, while the data for $\delta^{13}C_a$ are from SCRIPPS and fitted as shown in Fig. 3. The seasonal cycle amplitude of a given year is only computed if at least 10 monthly values are available. Number of years included in the trend calculation for $SA(C_a)$ and model-based $SA(\delta^{13}C_a)$ are given in parentheses. The observed trend for $C_a$ is affected by anomalous values at Key Biscayne and is not included. Over the period 1982–2012, significant trends (two-sided t-test at 5% significance) are only found for Alert, Barrow, Ocean Station M, and Mahe Island for observed $C_a$, for Mariana Islands, Mahe Island, Palmer, Halley, and South Pole for simulated $C_a$, and Ascension, Mahe Island, and South Pole for simulated $\delta^{13}C_a$. The decadal-scale trends are given per century for better readability.

| Site | Trend in Seasonal Cycle Amplitude | | | |
| --- | --- | --- | --- | --- |
| | $C_a$ [ppm/century] | | $\delta^{13}C_a$ [permil/century] | |
| | Observed | Standard | Observed | Standard (years) |
| Alert, Nunavut, Canada (82°N) | 6.5 ± 2.3 | 5.3 ± 2.7 | 0.1 ± 0.09 | 0.02 ± 0.13 ( 25 ) |
| Barrow, Alaska, US (71°N) | 9.5 ± 1.9 | 1.0 ± 3.6 | 0.14 ± 0.09 | -0.09 ± 0.15 ( 26 ) |
| Ocean Station M, Norw. (66°N) | 7.2 ± 3.1 | 7.7 ± 3.9 | | 0.02 ± 0.17 ( 25 ) |
| Cold Bay, Alaska, US (55°N) | 1.3 ± 4.4 | 5.2 ± 4.4 | | 0.10 ± 0.20 ( 25 ) |
| Shemya Island, Alaska, US (53°N) | -0.5 ± 5.3 | -0.4 ± 4.9 | | -0.04 ± 0.22 ( 21 ) |
| Mace Head, Ireland (53°N) | -9.6 ± 14.0 | -5.6 ± 4.3 | | -0.27 ± 0.24 ( 15 ) |
| Terceira Island, Portugal (39°N) | -1.5 ± 6.6 | 5.7 ± 3.7 | | 0.10 ± 0.17 ( 14 ) |
| Key Biscayne, Florida, US (26°N) | | 1.5 ± 1.7 | | 0.03 ± 0.07 ( 25 ) |
| Mauna Loa, Hawaii, US (20°N) | -1.6 ± 1.0 | 0.6 ± 1.4 | -0.08 ± 0.06 | -0.07 ± 0.05 ( 26 ) |
| Cape Kumukahi, Hawaii (20°N) | -2.1 ± 2.0 | 2.2 ± 1.6 | -0.09 ± 0.06 | -0.01 ± 0.06 ( 26 ) |
| Mariana Islands, Guam (13°N) | -2.6 ± 4.8 | 4.1 ± 1.5 | | 0.08 ± 0.05 ( 23 ) |
| Ragged Point, Barbados (13°N) | -2.1 ± 2.1 | -0.3 ± 1.7 | | -0.06 ± 0.07 ( 18 ) |
| Christmas Island, Kiribati (2°N) | -2.4 ± 1.8 | -1.9 ± 1.6 | -0.15 ± 0.05 | -0.13 ± 0.07 ( 19 ) |
| Ascension Island, UK (7.6 °S) | 2.7 ± 1.8 | -0.6 ± 1.4 | | -0.15 ± 0.05 ( 25 ) |
| Mahe Island, Seychelles (5°S) | 3.7 ± 1.4 | 6.4 ± 2.3 | | 0.30 ± 0.07 ( 13 ) |
| Tutuila, American Samoa (14°S) | 2.2 ± 1.3 | -0.5 ± 0.9 | -0.04 ± 0.06 | -0.04 ± 0.02 ( 26 ) |
| Palmer Station, Antarctica (65°S) | -0.4 ± 0.9 | 2.4 ± 1.1 | | 0.04 ± 0.04 ( 23 ) |
| Halley Station, Antarctica (76°S) | 0.1 ± 1.8 | 2.9 ± 1.1 | | 0.03 ± 0.04 ( 17 ) |
| South Pole, Antarctica (90°S) | 0.7 ± 0.8 | 2.8 ± 0.7 | -0.07 ± 0.03 | 0.06 ± 0.03 ( 26 ) |





hampered by interannual-to-decadal variability, short record lengths, and a small $SA(\delta^{13}C_a)$ in comparison to measurement uncertainty and variability as typical at Southern Hemisphere sites. For example, dividing $SA(\delta^{13}C_a)$ by two standard deviations of IAV yields a "signal-to-noise" ratio (Keller et al., 2014) below 2.7 at SH sites and as low as 1.1 at the South Pole and American Samoa (Table 1). Thus, $SA(\delta^{13}C_a$ would need to roughly double over the observational period for a trend in $SA(\delta^{13}C_a)$ to emerge from the noise of IAV at these two sites. The situation is more favourable for trend detection at NH extratropical sites (Table 1), where the signal-to-noise ratio ranges between 9 and 16 and changes of 6 to 11% in $SA(\delta^{13}C_a)$ would emerge. The Scripps data, including seasonality, are provided as (i) monthly samples, (ii) a fit to these monthly samples, and (iii) the monthly samples but missing values replaced with fitted values. We also used the original, non-gap-filled data and years with at least 9, 10, or 11 monthly values per year in the regression. Trends statistically different from zero are identified for CHR and SPO, whereas the trends at all other sites remain indistinguishable from zero (an exception is MLO when setting the data availability limit to 10 monthly values). For the fitted data, trends are statistically different from zero only at two sites (La Jolla and Christmas Island). In summary, observed temporal trends in $SA(\delta^{13}C_a)$ are small ($\leq 0.15$ permil/century) and not statistically different from zero (at p < 0.05) at individual sites A significant negative trend is found for the tropical site Christmas Island and detection of trends is difficult at the Southern Hemisphere sites, where $SA(\delta^{13}C_a)$ is small.

The $\delta^{13}C_a$ records of GLOBALVIEW-CO2C13 (2009) span order of a decade and are, therefore, not used for trend detection, although we evaluated trends from the simulations for the GLOBALVIEW sites (Table 2; excluding Key Biscayne). We require at least 10 monthly values for a year to be included in the linear regression. A limitation of our study is that transport matrices are only available for the period 1982 to 2012, limiting the analysis period and direct model-data comparison to three decades only. Simulated trends in $SA(\delta^{13}C_a)$ are small (often less than 0.01‰ per decade) and statistically insignificant (5% level) except at three SH sites (Ascencion, Mahe, South Pole), with a small seasonal cycle amplitude. Observed relative trends in $SA(C_a)$ are larger than in $SA(\delta^{13}C_a)$ at northern high latitudes and statistically significant at Alert, Point Barrow, Ocean Station, and Mahe Island but insignificant at all other sites over the 1982-2012 analysis period. Simulated trends in $SA(C_a)$ are insignificant, except at four SH sites and the Mariana Islands.

We compare model (m) and observed (o) slopes ($\beta$). Under the null hypothesis of no slope difference, the $T = (\beta_m - \beta_o)/\sqrt{s_{\beta_m}^2 + s_{\beta_o}^2}$ statistic (where $s_\beta$ is the standard error of the $\beta$ slope estimate) is Student t-distributed (Welch, 1947). Trends are different when the $T$ values are larger than the 0.975 quantile of a t-distribution with $\nu$ degrees of freedom (T>~2). Modeled and observed trends are different at one site, South Pole, for $SA(\delta^{13}C_a)$ and at one site (Barrow) for $SA(C_a)$. As will become clear in the next section, the largest surface-atmosphere isotope fluxes and temporal changes in these fluxes are simulated in the region north of $40^oN$. We are therefore interested in quantifying how well the model represents temporal changes in $SA(C_a)$ in this region and over a 40-year period, representative of the $\delta^{13}C_a$ observational record. For the five NH high-latitude sites with more than twenty years of data, uncertainties in the temporal change of $SA(C_a)$ range between 5 and 13% at individual sites over a 40-year period. The average trend in $SA(C_a)$ for these five NH sites (Alert, Barrow, Ocean Station M, Cold Bay, Shemmya Island) is 4.8±1.6 ppm/century (31±10 %/century) from observations and 3.8±1.8 ppm/century (22±11 %/century) from the model. The difference between and the uncertainty in these estimates translates into





a relative change in $SA(\mathrm{C}_a)$ of around 4% to 5% over a 40-year period. This suggests that our model chain represents well the observed temporal changes in $SA(\mathrm{C}_a)$ in the NH extratropical atmosphere.

Given the mostly insignificant trends at individual sites over the model analysis period 1982-2012, the question arises
whether larger trends are detected when considering longer time scales. Century scale trends, or their absence, can be readily estimated in the simulations by comparing $SA$ for the modern period (1982-2012) ($\mathrm{E_{standard}}$) and the control run under preindustrial conditions ($\mathrm{E_{control}}$) (Table 1; solid red versus dashed blue line in Fig. S3 and S4). For $\mathrm{C}_a$, a growth in $SA$ is clearly visible (12.2 ppm to 17.25 ppm at Alert; 6 ppm to 8.3 ppm at Mauna Loa; 1.7 to 2.1 ppm at the South Pole). Across all 19 sites, $SA(\mathrm{C}_a)$ has grown by 44% ± 35% (mean ± standard deviation) from 1700 AD to (1982-2010). The growth in
$SA(\mathrm{C}_a)$ ranges between 33% and 42% across the 12 extratropical NH sites (Table 1).

For $\delta^{13}\mathrm{C}_a$, the control simulation exhibits an almost identical $SA$ averaged across all 19 sites (2% ± 16% lower in $\mathrm{E_{control}}$ than $\mathrm{E_{standard}}$). Larger deviations are found at individual sites than on average (Fig. S4) with a moderately smaller $SA(\delta^{13}\mathrm{C}_a)$ in $\mathrm{E_{control}}$ at Alert, almost identical results are found at Mauna Loa and a slightly higher $SA(\delta^{13}\mathrm{C}_a)$ is found at the South Pole in $\mathrm{E_{control}}$ than $\mathrm{E_{standard}}$. The change in $SA(\delta^{13}\mathrm{C}_a)$ from preindustrial ($\mathrm{E_{control}}$) to modern ($\mathrm{E_{standard}}$) range between -6%
and 9% across the 12 extratropical NH sites, whereas more diverse results (-28% to + 63%) are simulated at the tropical and SH sites (Fig. S4, Table 1). The modelled industrial period change in $SA(\delta^{13}\mathrm{C}_a)$ does not emerge from the noise of variability, except at one tropical (Christmas) and three SH sites (Ascension, Mahe, Palmer). We applied the difference of $SA(\delta^{13}\mathrm{C}_a)$ between $\mathrm{E_{standard}}$ and $\mathrm{E_{control}}$ as signal $S$ and two standard deviations from IAV of $\mathrm{E_{standard}}$ (or from the observations) as noise $N$ and requiring $|S|/N > 1$ for the signal to emerge (Keller et al., 2014). The fact that trends in $SA(\mathrm{C}_a)$ and the near-zero
trends in $SA(\delta^{13}\mathrm{C}_a)$ are better identified by difference between the modern and preindustrial periods than by regression over the modern period motivates us to focus on the comparison between $\mathrm{E_{standard}}$ vs $\mathrm{E_{control}}$ in the remaining result sections.

### 4.3 Zonal decomposition of seasonal land-biosphere fluxes

#### 4.3.1 Changes in the seasonal amplitude of land-biosphere fluxes and $\delta^{13}\mathrm{C}_a$ over the historical period

Next, we address the near-absent temporal trends in $SA(\delta^{13}\mathrm{C}_a)$ at NH sites by analyzing the zonally-averaged cumulative
growing season flux of $|\delta^{13}f^*_{al,net}|$, i.e., $SA(\delta^{13}f^*_{al,net})$ (Fig. 4). The northern mid- to high-latitude ecosystem fluxes exhibit the largest seasonal cycle, followed by tropical rain-green forests and savannahs in $\mathrm{E_{standard}}$. This flux pattern contributes to the larger $SA(\delta^{13}\mathrm{C}_a)$ at the NH extratropics versus tropical and SH sites. A similar latitudinal flux pattern holds for $\mathrm{E_{control}}$.

Turning to the change over the historical period, $SA(\delta^{13}f^*_{al,net})$ is 28% larger for the region north of 15°N (30% larger for >40°N, and 20% larger for 15°N-40°N) for $\mathrm{E_{standard}}$ than $\mathrm{E_{control}}$. This growth is comparable to the observed increase
in atmospheric $CO_2$ of 32% from pre-industrial to the reference period of 1982-2012. In contrast, $\mathrm{E_{control}}$ sometimes exhibts larger $SA(\delta^{13}f^*_{al,net})$ than $\mathrm{E_{standard}}$ in the tropical and SH ecosystems (Fig. 4). Following Eq. 8, the near-proportional growth in the $SA(\delta^{13}f^*_{al,net})$ and annual mean atmospheric $CO_2$ in the NH extratropics is consistent with the absence of any major long-term change in $SA(\delta^{13}\mathrm{C}_a)$ at extratropical NH sites (Table 1 and Fig. S4). $SA(\delta^{13}\mathrm{C}_a)$ and its change at extratropical NH sites is dominated by the large $SA(\delta^{13}f^*_{al,net})$ in the northern extratropics (Fig. 4) and transport from low latitude regions is



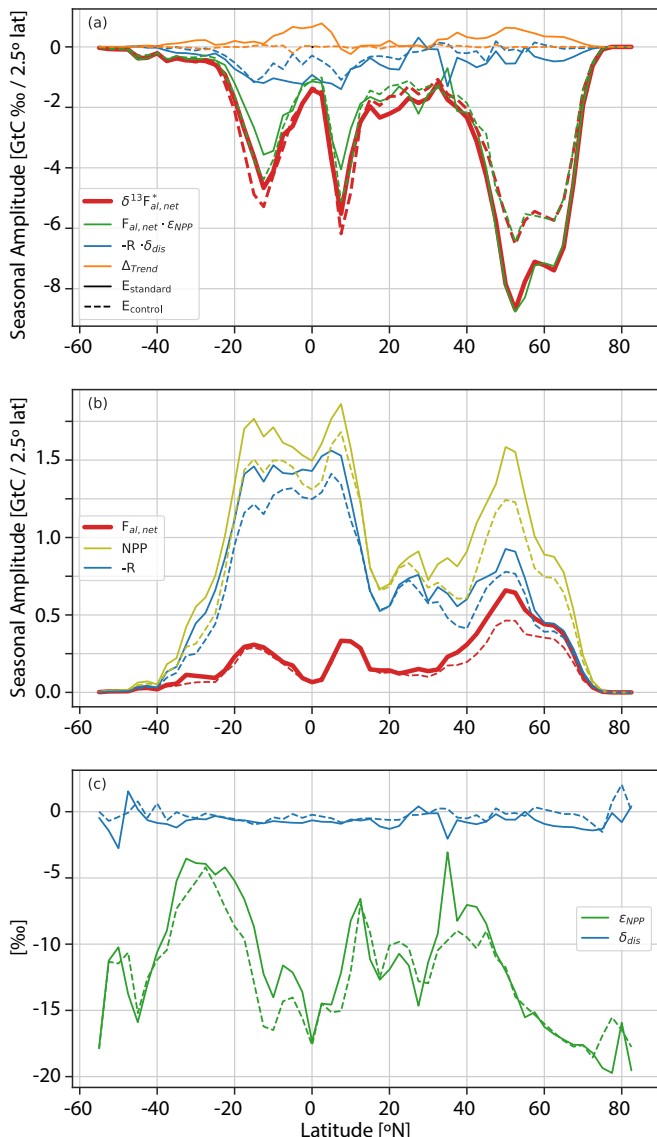

**Figure 4.** The seasonal amplitude per $2.5°$ latitude band of the signature-weighted, detrended net atmosphere-land flux $\delta^{13} f^*_{al,net}$ in the period 1982-2012 is shown in panel (a) in red. This quantity is the sum of three constituents seasonal amplitudes (Eq. 9 and Appendix A): Net land-atmosphere flux weighted with photosynthetic fractionation ($f_{al,net} \cdot \varepsilon_{NPP}$, green) plus release fluxes weighted with the disequilibrium signature ($R \cdot \delta_{dis,la}$, blue) plus the contribution to the seasonal amplitude by the underlying trend of $\delta^{13} f^*_{al,net}$ ($\Delta_{trend}$, orange) (sign convention: "green+blue+orange=red"). In panel (b), the seasonal amplitudes of (non-detrended) net carbon fluxes are shown. The net atmosphere-land flux ($f_{al,net}$ (red) is split in Net Primary Productivity (NPP, olive) and release flux ($R$, blue). In the bottom panel (c) the corresponding fractionation of photosynthesis $\varepsilon_{NPP}$ and the disequilibrium signature $\delta_{dis,la}$ is shown. All values are for the period with $\delta^{13} f^*_{al,net}$ smaller than zero ($\sim$ growing season). The y-axis in panel (a) is inverted to illustrate the anticorrelation of $\delta^{13}C$ and carbon fluxes. The results from the standard simulation ($E_{standard}$, solid lines) are compared to the preindustrial control simulation ($E_{control}$, dashed lines).





less important. On the other hand, the large extratropical $SA(\delta^{13}f^*_{al,net})$ influences $SA(\delta^{13}C_a)$ and its temporal changes at lower latitudes. Without this influence, we would, based on Eq. 8, expect a decrease in $SA(\delta^{13}C_a)$ outside the extratropics, given the relative increase in annual mean $CO_2$ is larger than the increase in $SA(\delta^{13}f^*_{al,net})$ in these regions.

For $CO_2$, the amplitude of the modeled zonally-averaged net atmosphere-to-land $CO_2$ flux, $SA(f_{al,net})$, shows the largest values in the NH extratropics and a large increase over the historical period of 33% in the region 15°N-90°N (15°N-40°N:
26% ; 40°N-90°N: 37%), driven by a larger increase in NPP than release fluxes ($R$), whereas $SA(f_{al,net})$ is smaller in the tropics and SH and shows hardly any changes from preindustrial (E$_{control}$) to modern (E$_{standard}$) south of 20°N.

### 4.3.2 The coupling between the seasonal amplitude of $C_a$ and $\delta^{13}C_a$

$SA(C_a)$ and $SA(\delta^{13}C_a)$ are partly coupled by the underlying carbon fluxes. The question arises to which extent $SA(\delta^{13}C_a)$ holds information independent from $SA(C_a)$. We decompose $\delta^{13}f^*_{al,net}$ in a contribution linked to the net atmosphere-to-land
carbon flux, $f_{al,net}$=NPP-$R$, and an isotopic disequilibrium flux (see Appendix A and section 3 for notation; $f_{al,net}$ is positive for a flux into the land biosphere):

$$\delta^{13}f^*_{al,net} = \underbrace{(\text{NPP}-R)}_{f_{al,net}} \cdot \varepsilon_{\text{NPP}} - R \cdot \underbrace{\left(\delta^{13}C_R - \delta^{13}C_{\text{NPP}}\right)}_{\delta_{dis,la}}. \tag{9}$$

NPP is the net primary productivity of all plants within a grid cell. $R$ is the sum of all land biosphere release fluxes to the atmosphere, such as those from heterotrophic respiration, fire, mortality, and product pools, except autotrophic respiration.
$\delta^{13}C_R$ is the signature of $R$ and $\delta^{13}C_{NPP}$ is the signature of NPP, with $\varepsilon_{\text{NPP}}$ (or $\varepsilon_{al}$) representing the (flux-weighted) fractionation by NPP. The difference in signatures of $R$ and NPP is the isotopic disequilibrium, $\delta_{dis,la}$. Eq. 9, together with Eqs. 7 and 8, provide insights into the driving factors for the seasonal amplitudes. Putting the ocean aside (Heimann et al., 1989), $SA(C_a)$ is driven by the spatio-temporal pattern of (NPP-$R$), whereas $SA(\delta^{13}C_a)$ is additionally influenced by seasonal variations in $\varepsilon_{\text{NPP}}$, and the disequilibrium flux ($R \cdot \delta_{dis,la}$).

The decomposition of zonally-averaged $SA(\delta^{13}f^*_{al,net})$ into the amplitude of constituent fluxes and their isotopic signatures is displayed in Figure 4 and Table S1. On the global average, $SA(f_{al,net} \cdot \varepsilon_{NPP})$, the amplitude of the net atmosphere-land flux ($f_{al,net}$) weighted with the signature of photosynthesis ($\varepsilon_{NPP}$), contributes with a fraction of 90% to $SA(\delta^{13}f^*_{al,net})$ for both E$_{standard}$ and E$_{control}$. For the region north of 40°N, $SA(-R \cdot \delta_{dis,la})$, the amplitude of the disequilibrium flux, contributes only 7% to $SA(\delta^{13}f^*_{al,net})$ in E$_{standard}$ and is almost negligible for E$_{control}$ (2%). This small contribution of the disequilibrium
flux ($-R \cdot \delta_{dis,la}$) relative to the net flux ($f_{al,net} \cdot \varepsilon_{NPP}$; Eq. 9) arises as the seasonal amplitude of the carbon release flux $R$ is similar in magnitude to that of the net land carbon uptake $f_{al,net}$ in the northern extratropics (blue vs red lines in Fig. 4b) while the disequilibrium $\delta_{dis,la}$ is an order of magnitude smaller than $\varepsilon_{NPP}$ (Fig. 4c). Thus in LPX, $SA(\delta^{13}C_a)$ is dominated by the growing season net carbon uptake flux in northern high-latitudes, suggesting that $SA(\delta^{13}C_a)$ holds little information on the isotopic disequilibrium at high latitude sites. In contrast, the contribution by the disequilibrium flux $SA(-R \cdot \delta_{dis,la})$ and
by the net carbon flux $SA(f_{al,net} \cdot \varepsilon_{NPP})$ are near-equal in the tropics (10°S-10°N) and the SH (Fig. 4, Table S1). Note that in E$_{control}$, $SA$ of the disequilibrium flux and the disequilibrium between photosynthesis and respiration (Eq. 9 and Fig. 4),





albeit smaller than in $E_{standard}$, are not negligible due to the lagged response of the respiration signatures to natural changes in $\varepsilon_{NPP}$. A small contribution ($\Delta_{trend}$) to the isotopic flux seasonality in $E_{standard}$ arises from the secular increase in flux (Fig. 4;see Appendix A). $SA$(NPP) and $SA(R)$ increase from preindustrial to modern not only in northern ecosystems but also in the tropics (Fig. 4b). $SA$(NPP-$R$) increases in the NH extratropics but does hardly change south of 20°N.

The zonal variation in (growing season) photosynthetic fractionation $\varepsilon_{NPP}$ is mainly due to differences in vegetation composition, with C4 plants having considerably lower photosynthetic fractionation than C3 plants (Fig. 4c). Land use and the evolving distribution of C3 and C4 crops are prescribed in the model and C4 grasses are more prevalent than C3 grasses in low-latitude dryland ecosystems. Accordingly, maxima in flux-weighted, zonal-mean $\varepsilon_{NPP}$ are simulated at 35°N, 12°N, and broadly around 30°S. Minima are simulated for the C3-dominated high-latitude ecosystems and tropical rain forest zone. In $E_{standard}$ $\varepsilon_{NPP}$ is generally less negative than in $E_{control}$ and increased by 1.18 ‰ (9% in relative units) on the global average ($SA$(NPP)-weighted) mainly due to the increase in the prevalence of C4 plants. To estimate the influence of the increase in C4 prevalence on global mean $\varepsilon_{NPP}$ (but not on global GPP), we run a factorial simulations, $E_{C3}$, with the fractionation formulation for all C4 plants replaced by those for C3 plants. The difference between $E_{standard}$ and $E_{C3}$, i.e., the change in fractionation attributable to C4 plants, amounts to about 1.5 ‰ on global average (1982-2012 versus 1720-1750) (Fig. S5). The disequilibrium signature $\delta_{dis,la}$ is closer to zero for the control than the standard simulation. This is expected because the signal from the atmospheric $^{13}$C Suess effect in $R$ is delayed relative to NPP by vegetation, soil, and product pool lifetimes, leading to a disequilibrium with respect to the production signature.

## 5 Discussion

### 5.1 Growth in seasonal cycle amplitudes: implications for stomatal conductance and water use

The seasonal amplitude of $CO_2$ ($SA(C_a)$) is observed to grow over time depending on location (Bacastow et al., 1985; Barlow et al., 2015; Piao et al., 2018) and driven by changes in the seasonality of net land carbon uptake (Graven et al., 2013; Forkel et al., 2016). The growth is captured by our model and simulated and observations-based trends generally agree within uncertainties at individual stations. Averaging the signals across northern high-latitude sites (>40°N), yields relatively small uncertainties in data and model slopes and their difference, corresponding to a change in seasonal amplitude of 5% (1-$\sigma$) over a 40-year period. Corresponding uncertainties and data-model mismatches in slopes are up to 21% for the individual high latitude sites. The good data-model agreement for the averaged trend suggests that our model chain broadly captures the influence of trends in the seasonal cycle of net land carbon uptake on trends in $SA(C_a)$ and $SA(\delta^{13}C_a)$ at northern sites.

Previous studies show northern ecosystems progressively taking up more carbon during the growing season (Graven et al., 2013; Forkel et al., 2016; Piao et al., 2018; Bastos et al., 2019). For example, Bastos et al. (2019) find a positive trend in $SA(f_{al,net})$ north of 40°N and small or no growth in $SA(f_{al,net})$ between 25°N and 40°N. Consistent with these studies, the seasonal cycle amplitude of NPP and the terrestrial release flux $R$ is simulated by LPX-Bern to increase over the industrial period in both hemispheres, but the growing season net carbon uptake ($SA$(NPP-R)=$SA(f_{al,net})$) is only increasing north of ~30°N (Fig. 4b).



For $\delta^{13}C_a$, the observations from the (GLOBALVIEW-CO2C13, 2009) and Scripps products do not show a statistically significant change of $SA(\delta^{13}C_a)$ at most sites with more than 20 years of data (Fig. 3). This is consistent with Gonsamo et al. (2017) who did not detect a temporal trend in $SA(\delta^{13}C_a)$ and seasonal phasing by fitting Scripps daily flask data from the four sites Alert, Point Barrow, La Jolla, and Mauna Loa. Modeled changes in $SA(\delta^{13}C_a)$ yield no clear trend in the standard case over the simulation period 1982 to 2012, consistent with the observations. Further, $SA(\delta^{13}C_a)$ is in general very similar for the
preindustrial control and the standard case at the NH sites (Table 1; dashed blue vs red line in Supplementary Fig. S4), again consistent with the statistically insignificant trend in the observations.

Theoretical considerations imply no trend in $SA(\delta^{13}C_a)$ for a proportional growth in the seasonality of net atmosphere-surface isotope flux ($\sim SA(\delta^{13}f^*_{al,net})$) and in background atmospheric $CO_2$ (see Eq. 8 section 3). Both parameters increased by about 30% over the industrial period over northern ecosystems (>40°N). The growth in $SA(\delta^{13}f^*_{al,net})$ is primarily driven
by the increase in growing season net carbon uptake ($SA(f_{al,net})$) at northern latitudes. We suggest that the near-proportional growth in $SA(\delta^{13}f^*_{al,net})$ and in background atmospheric $CO_2$ is mainly responsible for the statistically insignificant trend in $SA(\delta^{13}C_a)$ at high northern latitude sites, and contributing to the statistically insignificant trend in $SA(\delta^{13}C_a)$ at other NH sites via atmospheric transport.

Factorial simulations, with an individual forcing kept at preindustrial, show small individual contributions by climate change,
fossil emissions, and land use to the industrial period growth in $SA(\delta^{13}C_a)$ at northern extratropical sites (Fig. S3 and S4). This suggests that the statistically insignificant trend in $SA(\delta^{13}C_a)$ at northern extratropical sites is not caused by offsetting impacts of climate change versus increasing $CO_2$. Fossil fuel emissions cause an increase and land use change a reduction in $SA(\delta^{13}C_a)$ at low latitude and southern sites (Fig. S3 and S4). We attribute the dampening influence of land use change to the replacement of C4 plants by C3 crops causing a general shift in the fractionation during photosynthesis to less negative values
south of $\sim$45°N (Fig. 4c). This damping influence highlights the importance of considering spatiotemporal variations in C3 and C4 plant distributions when analyzing $\delta^{13}C_a$.

Our results hold implications for carbon and water fluxes, and evaporative cooling. The good agreement between observations and model results for $SA(\delta^{13}C_a)$ and its temporal trend provides implicit support for regulation of stomatal conductance by C3 plants towards a constant ratio of the $CO_2$ mole fraction in the leaf intercellular space ($c_i$) and ambient atmospheric air
($c_a$) on the continental scale. Following Farquhar (1989), the fractionation for C3 photosynthesis and NPP ($\varepsilon_{\mathrm{NPP}}$) is proportional to $c_i/c_a$:

$$\varepsilon_{\mathrm{NPP}} = a + (b-a) \cdot \frac{c_i}{c_a}, \tag{10}$$

with $a$ (4.4), and $b$ (27) being constants. Two contrasting scenarios are published for the regulation of leaf stomatal conductance for C3 plants. First, many site studies (Voelker et al., 2016; Saurer et al., 2014; Kauwe et al., 2013; Peñuelas et al., 2011; Frank
et al., 2015; Keller et al, 2017) suggest a regulation of stomatal conductance towards a constant $c_i/c_a$ and, hence, $c_i$ to grow proportional to $c_a$. An absent temporal trend in $c_i/c_a$ translates into an absent trend in $\varepsilon_{\mathrm{NPP}}$, and vice versa (Eq. 10). Focusing on regions north of >40°N, where carbon fluxes are largest and C3 plants dominate, LPX-Bern simulates a small role of isotopic disequilibrium fluxes and a dominant influence of net atmosphere-surface fluxes on $SA(\delta^{13}C_a)$ (Fig. 4, green vs





blue lines). Importantly, LPX-Bern simulates small temporal changes in the (flux-weighted) fractionation of the zonally and

seasonally integrated NPP at northern sites (Fig. 4c, green lines) and a stomatal regulation towards constant $c_i/c_a$. In turn, the good model-data agreement in the temporal trends of $SA(\mathrm{C}_a)$ and $SA(\delta^{13}\mathrm{C}_a)$ imply consistency with the observational evidence for this scenario towards constant $c_i/c_a$.

In contrast, Battipaglia et al. (2013) and Keenan et al. (2013) suggest a regulation of stomatal conductance towards a constant $c_i$ and a decreasing ratio $c_i/c_a$ under rising CO$_2$. Evaluating Eqs. 10 for 1980-2022, the period with $\delta^{13}\mathrm{C}_a$ measurements,

yields a decrease in $\varepsilon_{\mathrm{NPP}}$ of 15% (-3.0 to -3.8 ‰) for an initial $c_i/c_a$ ratio in the range of 0.7 to 0.9 and constant $c_i$. We argue that the good observation-model agreement in the simulated trends in $SA(\mathrm{C}_a)$ implies that the influence of the simulated net atmosphere-land carbon flux is realistic and $SA(\delta^{13}\mathrm{C}_a)$ would decrease if $\varepsilon_{\mathrm{NPP}}$ decreases. A decrease in $SA(\delta^{13}\mathrm{C}_a)$ of 15% would emerge from the noise of variability at individual northern sites. Taken together, we suggest that the scenario towards constant $c_i/c_a$ is consistent with the observations whereas the scenario towards constant $c_i$ appears less likely. However,

uncertainties remain and our conclusions for the two scenarios of stomatal regulation await confirmation by other modelling studies.

The two scenarios imply large differences in water fluxes (Knauer et al., 2017). The intrinsic water use efficiency (iWUE), the ratio between assimilation of CO$_2$ by photosynthesis ($A$) and conductance of CO$_2$ ($g$), is, as $\varepsilon_{\mathrm{NPP}}$, a function of $c_i$ and $c_a$:

$$iWUE = \frac{A}{g} = c_a \cdot \left(1 - \frac{c_i}{c_a}\right). \tag{11}$$

iWUE would have increased from 1980 to 2022 by 23% for $c_i/c_a$ constant, but by 77 to 231% for $c_i$ constant, assuming an initial $c_i/c_a$ of 0.7 to 0.9. In the latter scenario, stomatal conductance and, correspondingly, water loss per stomatal pore, would have decreased strongly over the last decades.

## 5.2  Seasonality: C$_a$ versus $\delta^{13}$C$_a$ and isotopic disequilibrium

Observations of $\delta^{13}\mathrm{C}_a$ seasonality provide different, complementary information compared to observations of C$_a$ seasonality.

As discussed above, the model results suggest that the additional information of $SA(\delta^{13}\mathrm{C}_a)$ compared to $SA(\mathrm{C}_a)$ is in the magnitude of $\varepsilon_{\mathrm{NPP}}$ at northern high-latitude sites. The contribution of the isotopic disequilibrium flux, indicative of the transit time of carbon through the land biosphere, plays a relatively small role in the northern extratropics, given the large growing season net carbon uptake flux in this region (Fig. 4). In contrast, for tropical and SH ecosystems, the terrestrial isotopic disequilibrium flux plays a large role (blue versus green line in Fig. 4a), at least in our model. However, the seasonality

of $\delta^{13}\mathrm{C}_a$ and C$_a$ at the tropical background monitoring sites analyzed in this study is strongly influenced by long-range transport, adding uncertainty to the interpretation of seasonal signals. Further, we have limited understanding of net ecosystem exchange in the tropics, and uptake and release fluxes may cancel largely in the evergreen rain forest limiting the value of seasonal analyses in this biome. Monitoring C$_a$ and $\delta^{13}\mathrm{C}_a$ over tropical and SH land regions could potentially provide valid information to disentangle NPP, respiration, and net carbon fluxes. Ideally, seasonally-resolved observations are taken in air

masses influenced primarily by regional land biosphere fluxes, thereby minimizing uncertainties from long-range transport, and interpreted with the help of atmospheric transport and land biosphere models (Botía et al., 2022). For example, the data



may be assimilated into atmospheric transport models applied in inverse mode to infer surface carbon and isotope fluxes or into isotope-enabled land biosphere models, combined with atmospheric transport, to optimize parameters governing modelled carbon and isotope fluxes (Peylin et al., 2016; Castro-Morales et al., 2019).

While the influence of the gross exchange flux and the isotopic disequilibrium on $\delta^{13}C_a$ seasonality is modeled to be small at northern sites for today, it remains to be explored how global warming will change these parameters, e.g., due to changes in fire frequency and tree mortality, and affect $\delta^{13}C_a$ and the information provided by continued $\delta^{13}C_a$ observations. We may also expect different disequilibrium fluxes and, in turn, $\delta^{13}C_a$ seasonality if the global carbon sink is driven by a stimulation of NPP, e.g., by $CO_2$ fertilization (Walker et al., 2021) as in LPX-Bern, versus a change in tree longevity (Bugmann and Christof, 2011; Körner, 2017). It remains to investigate, e.g., by applying perturbed parameter ensembles and sensitivity simulations, whether such differences indeed significantly affect $\delta^{13}C_a$ seasonality.

The role of isotopic disequilibrium fluxes depends on the timescales considered. Analyses of the global budget of annual-mean atmospheric $\delta^{13}C(CO_2)$ (Keeling et al., 1989; Francey et al., 1995; Joos and Bruno, 1998; Trudinger et al., 2002; van der Velde et al., 2013) indicate that the emissions of isotopically-light carbon from fossil fuel burning are primarily balanced by a reduction in the atmospheric inventory and ocean and land disequilibrium fluxes, leaving a relatively small role for the isotopic flux associated with the net ocean and land carbon sink. Ocean and land fluxes are about equally important for the removal of the fossil fuel $\delta^{13}C(C_a)$ perturbation. For the seasonality of $\delta^{13}C(C_a)$, our results confirm that the land fluxes exert a dominant influence compared to ocean fluxes at NH sites and we find, on the global average, that the disequilibrium flux contributes a small fraction to the seasonal amplitude of the net land isotope flux.

On a technical note, transporting simulated $^{13}C$ fluxes is not without challenges. The definition of the $\delta$-notation can pose numerical difficulties when net $^{12}C$ fluxes are close to zero. We find that transporting signature-weighted total carbon fluxes is the most reliable method for arriving at local $\delta^{13}C_a$. Similarly, seemingly small errors in the model representation of gross fluxes and mass balances, can become critical when considering net surface-to-atmosphere fluxes.

## 6    Conclusions

In conclusion, we explored the global-scale mechanisms driving the observed seasonal cycle of atmospheric $\delta^{13}C(CO_2)$ and $CO_2$ at 19 monitoring sites using atmosphere-surface fluxes from the Bern3D-LPX Earth System Model of Intermediate Complexity and fossil emissions in combination with transport matrices from the TM3 atmospheric transport model. We find good data-model agreement at northern and tropical sites. No significant trends are detected nor modeled in the seasonal cycle amplitude of $\delta^{13}C(CO_2)$ at most monitoring sites, in contrast to the positive trends in the seasonal amplitude of $CO_2$. We attribute the statistically insignificant trend in the seasonal amplitude of $\delta^{13}(CO_2)$ to a near-equal percentage increase in the growing season net carbon uptake and isotope flux and the background atmospheric $CO_2$ in the northern extratropical land regions. Over the industrial period and at low-latitude and SH sites, land use change has a dampening influence on $\delta^{13}C_a$ seasonality through the replacement of C3 plants by C4 crops. Modeled isotopic disequilibrium fluxes have a small influence on the seasonal signal of $\delta^{13}C(CO_2)$ at NH sites, but play an important role in tropic and SH ecosystems, suggesting that monitoring



the $\delta^{13}C_a$ seasonality over tropical and SH land would provide valuable information on gross carbon exchange fluxes and the time scales of carbon turnover in the land biosphere. Our results, based on a single model chain, provide implicit support for a regulation of the stomatal conductance of C3 plants towards a constant $c_i/c_a$ on biome scales and intrinsic water use efficiency to grow proportionally to atmospheric $CO_2$ over recent decades with implications for carbon and water fluxes. More generally, the results suggest that observations of the $\delta^{13}C_a$ seasonal cycle offer highly useful information on carbon and water cycle processes. We recommend to apply seasonally-resolved $\delta^{13}C_a$ observations as a novel constraint for land biosphere models used to simulate the terrestrial sink of anthropogenic carbon and land use emissions. Future studies may employ an ensemble of isotope enabled models and perturbed parameter ensembles to elucidate whether our findings are robust and which models or process assumptions are compatible or incompatible with $\delta^{13}C_a$ data for improved projections of atmospheric $CO_2$ and global warming.

*Code and data availability.* The data from the Scripps $CO_2$ program are available here: https://scrippsco2.ucsd.edu/data/atmospheric_co2/. The GLOBALVIEW data from the Global Monitoring Laboratory were downloaded here: https://gml.noaa.gov/ccgg/globalview/. The data displayed in the Figures will be made freely available at Zenodo or a smilar site when the manuscript is accepted. For the review process the data and plotting scripts are available as a download: https://cloud.climate.unibe.ch/s/g9qrit7KDRnrbLp

## Appendix A: Decomposition of $\delta^{13} f^*_{as,net}$ and the calculation of seasonal amplitudes

We reformulate the net isotope flux in terms of net and gross carbon fluxes, isotopic fractionation, and isotopic disequilibrium (e.g., Mook (1986); Joos and Bruno (1998)) to diagnose their influence on the seasonal cycles.

The fractionation for a gross flux, e.g., from the atmosphere to the surface, is:

$$\varepsilon_{as} \cong (\delta^{13}C_{as} - \delta^{13}C_a), \tag{A1}$$

with $\delta^{13}C_{as}$ the signature of the gross flux from $a$ to $s$ ($f_{as}$) and $\delta^{13}C_a$ the signature of the source. The isotopic disequilibrium (or difference) between atmosphere-surface gross fluxes is:

$$\delta_{dis,sa} = -\delta_{dis,as} = \left(\delta^{13}C_{sa} - \delta^{13}C_{as}\right) \tag{A2}$$

The net carbon and isotope fluxes are differences between gross fluxes:

$$f_{as,net} = f_{as} - f_{sa} \tag{A3}$$

$$\delta^{13} f^*_{as,net} = f_{as} \cdot \left(\delta^{13}C_{as} - \delta^{13}C_a\right) - f_{sa} \cdot \left(\delta^{13}C_{sa} - \delta^{13}C_a\right) \tag{A4}$$

Rearranging yields:

$$\delta^{13} f^*_{as,net} = f_{as,net} \cdot \underbrace{\left(\delta^{13}C_{as} - \delta^{13}C_a\right)}_{\varepsilon_{as}} - f_{sa} \cdot \underbrace{\left(\delta^{13}C_{sa} - \delta^{13}C_{as}\right)}_{\delta_{dis,sa}} \tag{A5}$$




For the land biosphere (index $l$), it follows from Eqs. A3 and A5:

$$f_{al,net} = \text{NPP} - R \tag{A6}$$


$$\delta^{13} f^*_{al,net} = f_{al,net} \cdot \varepsilon_{\text{NPP}} - R \cdot \delta_{dis,la}, \tag{A7}$$

with:

$$\delta_{dis,la} = \delta^{13}\text{C}_R - \delta^{13}\text{C}_{\text{NPP}} \tag{A8}$$

NPP is the net primary productivity of all plants within a grid cell. $R$ is the sum of all release fluxes to the atmosphere, such

as those from heterotrophic respiration, fire, mortality, and product pools. $\delta^{13}\text{C}_R$ is the signature of $R$ and $\delta^{13}\text{C}_{NPP}$ is the signature of NPP, with $\varepsilon_{\text{NPP}}$ (or $\varepsilon_{al}$) representing the (flux-weighted) fractionation by NPP. Here, as in LPX-Bern, we have assumed that the uptake difference between gross primary production (GPP) and NPP is released on short time scales and without further carbon isotope fractionation.

The seasonal amplitudes of $\delta^{13} f^*_{al,net}$ and its components are calculated as follows. The time series of $\delta^{13} f^*_{al,net}$ is detrended

and normalized to zero. The trend is computed by a rolling 12-month mean of $\delta^{13} f^*_{al,net}$. Then, the resulting trend curve is subtracted from $\delta^{13} f^*_{al,net}$ (disregarding the first and last 6 months of the original series) to get a detrended curve. Finally, the detrended curve is normalized by subtracting its period mean. $\Delta_{trend}$ (e.g., in units of mol permil $\text{yr}^{-1}$ $\text{m}^{-2}$) is the difference between $\delta^{13} f^*_{al,net}$ after and before this detrending and normalizing procedure. We define a seasonal mask to compute seasonal amplitudes of fluxes and their signatures. For each model year, we identify months in which detrended $\delta^{13} f^*_{al,net}$ is negative or

equal to zero (roughly corresponding to the growing season). The sum of fluxes of these months is then termed the "seasonal amplitude" in a given year. For $\delta^{13} f^*_{al,net}$, this procedure is consistent with considering the difference between maximum and minimum values of the detrended cumulative sum of $\delta^{13} f^*_{al,net}$. Accordingly, the seasonal amplitudes of the component fluxes contributing to $\delta^{13} f^*_{al,net}$ are computed by summation over months where $\delta^{13} f^*_{al,net}$ is less or equal to zero within a given year. Component fluxes are [(NPP$-R$) $\cdot \varepsilon_{\text{NPP}}$], [$R \cdot \delta_{dis,la}$], [$\Delta_{trend}$], and, further, [NPP], [$R$], and [NPP-$R$] (Fig. 4). These

component fluxes are not detrended to readily calculate the signatures $\delta_{dis,la}$ and $\varepsilon_{NPP}$ by division of the seasonal amplitude isotopic flux with the corresponding seasonal amplitude carbon flux.

We note that the annual climatological mean values of the isotopic disequilibrium ($\delta_{dis,la}$), the net carbon flux ($f_{al,net}$), and the net isotopic flux ($\delta^{13} f^*_{al,net}$) vanish by definition for the preindustrial equilibrium. However, this does not hold for their seasonal amplitudes. Further, detrending $\delta^{13} f^*_{al,net}$ before the computation of its seasonal amplitude is consistent with

the calculation of the $\text{C}_a$ and $\delta^{13}\text{C}_a$ seasonal amplitude from the detrended atmospheric time series.

The seasonal cycles of $\text{C}_a$ or $\delta^{13}\text{C}_a$ are computed from observations and the TM3 results using the following procedure for either $\text{C}_a$ or $\delta^{13}\text{C}_a$, respectively. Months with missing values in either the observation or the TM3 simulation are masked in the TM3 and observational time series. Then the time series are detrended using a 12-month rolling mean and the overall mean of the series is set to zero to get for year, $y$, and month, $m$, seasonal anomalies $\Delta\text{C}_a(y,m)$ and $\Delta\delta^{13}\text{C}_a(y,m)$. Finally, the



period means for each calendar month, $\overline{\Delta C_a(m)}$ and $\overline{\Delta\delta^{13}C_a(m)}$, are computed by averaging over all corresponding monthly values. Additionally, the standard deviation is computed for each calendar month to inform about the interannual variability of the seasonality. The period-mean $SA$ is computed as the difference between the month with the highest ($\overline{m_{\max}}$) and lowest ($\overline{m_{\min}}$) value in $\overline{\Delta C_a(m)}$ and $\overline{\Delta\delta^{13}C_a(m)}$, respectively. For individual years, we computed $SA$ by difference from the extreme monthly values of each year.

*Author contributions.* FJ and SL wrote the manuscript with inputs from SZ. SL performed all model runs and FJ the statistical analyses. SL and FJ produced the figures and tables.

*Competing interests.* The authors declare to have no competing interests

*Acknowledgements.* This project has received funding from the European Union's Horizon 2020 research and innovation programme under grant agreement No 821003 (project 4C, Climate-Carbon Interactions in the Current Century) and by the Swiss National Science Foundation

(project #200020_200511). The work reflects only the authors' view; the European Commission and their executive agency are not responsible for any use that may be made of the information the work contains. We thank the researchers of the Cooperative Atmospheric Data Integration Project, NOAA ESRL, Boulder, Colorado, and of the Scripps $CO_2$ program for making their $CO_2$ and $\delta^{13}C$ data freely available and Martin Heimann for suggesting to plot seasonal anomalies of $CO_2$ versus those of $\delta^{13}C$. A special thanks goes to Christoph Köstler for providing the TM3 transport matrices and to Aurich Jeltsch-Thömmes for help with Bern3D.



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
