# Peer review of "No increase is detected and modeled for the seasonal cycle amplitude of $\delta^{13}{\rm C}$ of atmospheric carbon dioxide"

_EGUsphere, 2024_

## Referee Comment (RC1)

No increase is detected and modelled for the seasonal cycle amplitude of $\delta^{13}$C of atmospheric carbon dioxide

Fortunat Joos et al,

I.   General

This paper showed the result of stable isotope ($\delta^{13}$C) in $CO_2$ and $CO_2$ seasonal variation and its amplitude changes for a long-term period through observation and model investigations. The model and observation showed no significant changes of seasonal amplitude of $\delta^{13}$C while those values are increasing of atmospheric $CO_2$. Authors tried to understand why they showed different characteristics using possible tools and explain it. This is interesting and valued paper to understand carbon cycle and to lead readers why we monitor not only atmospheric $CO_2$ but also $\delta^{13}$C. However, for readers, it is hard to say this is well written so that a revision is necessary before publishing. The key revision is for clarification.

1. Some of explanations should be included to methods part rather than result or discussion section. Those explanations can make readers not focus on the main result.
2. There are many abbreviations without full names in the manuscript.
3. Once authors defined a term, please keep the defined term in whole manuscript (e.g., such as $C_a$ and $SA(C_a)$).
4. It might be good to reconsider whether the title is representative of whole manuscript. Authors would like to emphasize no significant changes of $\delta^{13}$C seasonal amplitude; however, did not mention about period (a decade or -100 years?), scale (global or Northern Hemisphere or Europe?) and the tool (only model or both of model and observation?). Also the no significant $\delta^{13}$C seasonal amplitude trend can be a trigger to investigate this experiment though, I wonder it can be a title of the manuscript.
5. It seems like very vague of function of result and discussion section. Normally when authors divide into two sections, result section should include only the experimental result which are from suggested method section and explain reasons why experiment show the result in discussion section. In this manuscript, even in result section, the reasons of experimental result were partly discussed which seems like very similar function of discussion section. And also, to discuss the result, in discussion section summary of results was suggested once again. This makes the manuscript very long and not clear. Hope authors reconsider the structure of manuscript and find out effective way to deliver what this paper really would like to say. One of ways is to combine two sections. I would like to suggest good example with good structure, Piao et al.,2018. Please consider the structure of the manuscript.

II. Specific
1. Title can be reconsidered.
2. L5: Though the authors have done an experiment on global scale, the word "to simulate **local** atmospheric $\delta^{13}C(CO_2)$" can make readers misunderstand that this result can have a bias from scale differences. It would be good to mention just "atmospheric $\delta^{13}C(CO_2)$" minus local or put more appropriate word in place of "local" (for example global background stations?)
3. L24: It might be wonder $\delta^{13}C$ is same to $\delta^{13}C_a$ in Line 20. If it is same, please revise and unify all terms in the manuscript. e.g. L27 as well.
4. L25: Recommend that authors can suggest definitions of $\delta^{13}C$ before mentioning $^{13}C$ and $^{12}C$, especially if $\delta^{13}C$ was used differently from $\delta^{13}C_a$ (for example: $^{13}C/^{12}C$, normally expressed relative to a standard as $\delta^{13}C(CO_2)$ in units of per mille (‰)).
5. L78: 19 sites represent of global levels? To avoid the question of error of this experiment, the reason to choose 19 sites can be discussed somewhere in section 2.3 and here explicitly mention as 19 global sites.
6. L82: This sentence is not suitable for introduction. Introduction is not abstract. This is one of results in this manuscript. Hope it can be moved to another section or removed.

   *We demonstrate for the first time that the observations at the globally distributed sites show no significant trends in the seasonal cycle amplitude of δ13Ca, consistent with our model chain, but surprising in view of the large trend in the seasonal amplitude of CO2.*

7. Section 2.1. There are too many abbreviations without full name. L92 (EMIC Bern3D-LPX), L93(Bern3D), L94(LPX), L98(DIC), L18 (LUH2 and NMIP), L120 (NCEP/NCAR), L121 (CRU-TS4.05).
8. L123: agents to species.
9. L123: the specific definition of $E_{control}$.
10. L125: Atmospheric $CO_2$ to Ca, $\delta^{13}C$ to $\delta^{13}C_a$.
11. L127: what does TM3 stand for?
12. L129 and L130: Just write Ca and $\delta^{13}C_a$ because they were defined already in previous section.
13. L135: $^{13}C$ or $\delta^{13}C_a$ ? Those confusions occurred all manuscript.
14. L140: Ca and $\delta^{13}C_a$? If not, I think those are also redefined as similar format. For example, $\delta^{13}C_o$ (ocean) or $\delta^{13}C_o$ (observation) etc. This is similar to the L142 and 143.
15. L152 to 153: The authors should mention the URL with last access date from Cooperative Global Atmospheric Data Integration Project and Scripps $CO_2$ program. I highly recommend adding that information. This is very important part.
16. Section 2.3: Authors should explain the reason why 19 sites were selected for the experiment and their measurement uncertainty. Also NOAA data and Scripps data have different scale and there is a bias between two data derived from two scales (Lueker et al., 2020)

17. L160: If authors did not use KER and NZD data, it would be good to not discuss here to avoid confusions.
18. L162: Hope you can keep the same term all over the manuscript, for Ca and $\delta^{13}$Ca.
19. L166: $^{13}$C to $\delta^{13}$Ca?
20. L191: Background $CO_2$ mixing ratio is different from observed $CO_2$ mixing ratio. This is important part because just observation $CO_2$ include local signals but background $CO_2$ is selected representative values from all observation data. Authors should discuss this in data section 2.3 as indicating which data were used for the experiment. Also please do not use $CO_2$ mixing ratio in place of Ca. If authors defined Ca, please keep the term. Also the unit of data that are used for this paper are not mixing ratio, that is mole fraction (Green book, 2007, Note. Official name is not green book but normally use as green book.)
21. L206: The model…, there are many models in the method section. It would be good to indicate explicitly what model is. For L212 The land biosphere model and L216 The ocean model should be explained for what kind of models were used, as well. Also, it would be good to match all experiment results with 2. Method section.
22. L244: Again, $\delta^{13}$C and $CO_2$ are differed from Ca and $\delta^{13}$Ca?
23. Table 1: It would be helpful to display the stations according to the latitude. Maybe swap Mahe Island and Acension Island?
24. L227: Does "from 1982 to 2012" mean Estandard? Keep the term in whole manuscript.
25. L234: Does "Standard simulation" mean Estandard?
26. L254: South Pole, Palmer, and Halley, L255: Figs.2, S1, and S2.
27. Figure 3. It would be good to add latitude information next to the name of stations. Also next to Panel (a), 'Data from Scripps…' can be removed.
28. L284: the unit 'permil/century' is difficult to understand through Fig.3.
29. Table 2: Can they be revised?; Observation data $CO_2$ to observation $C_a$, GlOBALVIEW-CO2 to $CO_2$, SCRIPPS to Scripps.
30. L291: SA($\delta^{13}$Ca)
31. L294: This can be moved to Method section.

The Scripps data, including seasonality, are provided as (i) monthly samples, (ii) a fit to these monthly samples, and (iii) the monthly samples but missing values replaced with fitted values. We also used the original, non-gap-filled data and years with at least 9, 10, or 11 monthly values per year in the regression

32. L311: Why do authors analyze model and observation slope? Please add the purpose.
33. L326: SA means SA($C_a$) or SA($\delta^{13}$C$_a$)? Or both of SA?
34. L335: Authors mentioned only the diverse range of SA, but the values seem like very significant. The explanations are focused on they are reliable data rather than the meaning of values. Was this discussed somewhere in the manuscript?
35. L336: Does 'industrial period' mean Estandard?

36. L350: Does 'pre-industrial to the reference period' mean that '$E_{control}$ to $E_{standard}$'?
37. L363 to L374: Can we move whole part to Method section? Or combine to Appendix A?
38. L406 to L441: Those are explained already in Section 4 and more similar to summary rather than discussion section. Only differences are adding references more. It would be good to make it simple and clear as suggested general review.
39. L409: What is the number of 'relatively small uncertainties'? and for 'no clear trend in the standard case' in L423.
40. L423, L425: If 'standard case' and 'preindustrial control' mean that $E_{control}$ and $E_{standard}$, please keep the same therm.
41. L445: NPP is different from the NPP in section 4.3.2? If same, why do authors invite another term, $\varepsilon_{NPP}$, here?
42. Section 5.2: I have quite similar opinion to section 5.1. The manuscript was mixed with result (discussed before) and seems like more conclusion section? It is very vague what authors really would like to say.

Reference

Piao et al.(2017) https://doi.org/10.1111/gcb.13909
Lueker et al., (2020) https://escholarship.org/uc/item/4n93p288

Greenbook(2007)          https://iupac.org/wp-content/uploads/2019/05/IUPAC-GB3-2012-2ndPrinting-PDFsearchable.pdf

---

## Author Response (AR1)

**Reply to the review comments for MS egusphere-2024-1972: No increase is detected and modeled for the seasonal cycle amplitude of δ13C of atmospheric carbon dioxide**

We thank the editor for handling the manuscript and both reviewers and the editor for their careful review and valuable comments that helped us to improve the presentation of our results. The original comments are given in black and our answer in blue fonts. Line numbers refer to the originally submitted version. A revised version of the manuscript with changes highlighted is added at the end of the reply.

Following the advice of the reviewers, we have shortened the discussion and restructured the manuscript by combining the results and discussion section. Our scientific results and conclusions remain unchanged.

**Comments by the editor**

Dear Dr. Joos,

Thank you for providing detailed responses to the comments and suggestions offered by the two reviewers.

Both reviewers recognized the significance of your work and recommended that the manuscript be published after some minor revisions. Based on the positive evaluations of the two reviewers and my perusal, I recommend 'Publish subject to minor revisions (review by editor)'. Please note that editors provide the final decision after authors have submitted a revised manuscript in response to this initial decision. Please also consider the following editorial suggestions:
Thank you for advise and your positive decision.

- δ13C(CO2): though this might be acceptable in your specific field, it would be more reader-friendly, if you provided a more widely used form (e.g., δ13C in CO2) or some definition at its first use.
Done. We replaced "$\delta^{13}C(CO_2)$" with "$\delta^{13}C$ of atmospheric $CO_2$ ($\delta^{13}C_a$)" at L1 of the abstract and correspondingly at line 489 of the conclusion. We replaced all remaining occurrences of the term "$\delta^{13}C(CO_2)$" with "$\delta^{13}C_a$" (6 replacements in the abstract).
- Line 6 "is detected": a past tense would be preferable if the sentence describes an observation.
Done. Changed "is detected" to "was detected"
- Lines 7-9 (and at the first use of these terms in the main text): Please specify the periods for both "preindustrial and modern periods" and this study.
Done. Text reads now: "Comparing the preindustrial (1700) and modern (1982-2012) periods, the modelled .."

When you have completed revising the manuscript, I ask you to make all the changes easily identifiable in a marked-up manuscript based on your point-by-point responses to the reviewers' comments. If possible, please specify the line numbers of the revised parts in your final responses accompanying the revised manuscript

Sincerely,

Ji-Hyung Park
Associate Editor, Biogeosciences

**RC1**

    I.        General

This paper showed the result of stable isotope (δ13C) in CO2 and CO2 seasonal variation and its amplitude changes for a long-term period through observation and model investigations. The model and observation showed no significant changes of seasonal amplitude of δ13C while those values are increasing of atmospheric CO2. Authors tried to understand why they showed different characteristics using possible tools and explain it. This is interesting and valued paper to understand carbon cycle and to lead readers why we monitor not only atmospheric CO2 but also δ13C. However, for readers, it is hard to say this is well written so that a revision is necessary before publishing. The key revision is for clarification.

Thank you for your support, the careful review, and your request for clarifications that helped us to improve the presentation of the manuscript. In response to the comments by both reviewers we removed the discussion section and incorporated part of the content in earlier sections.

1. Some of explanations should be included to methods part rather than result or discussion section. Those explanations can make readers not focus on the main result.
   We have moved text to the method sections. Please see our answer to your specific comments.

2. There are many abbreviations without full names in the manuscript.
   We added explanations of the abbreviations. Please see our answer to your specific comments.

3. Once authors defined a term, please keep the defined term in whole manuscript (e.g., such as Ca and SA(Ca)).
   We now use adopted terms throughout the MS as suggested.

4. It might be good to reconsider whether the title is representative of whole manuscript. Authors would like to emphasize no significant changes of δ13C seasonal amplitude; however, did not mention about period (a decade or -100 years?), scale (global or Northern Hemisphere or Europe?) and the tool (only model or both of model and observation?). Also the no significant δ13C seasonal amplitude trend can be a trigger to investigate this experiment though, I wonder it can be a title of the manuscript.
   We prefer to keep a simple title following the example of Piao et al., GCB, 2017. These authors used the title "On the causes of trends in the seasonal amplitude of atmospheric $CO_2$" for a publication discussing observed and modelled trends in $CO_2$ seasonality at northern hemisphere sites. The word "detected and modelled" in our title point the reader to observations and models.

5. It seems like very vague of function of result and discussion section. Normally when authors divide into two sections, result section should include only the experimental result which are from suggested method section and explain reasons why experiment show the result in discussion section. In this manuscript, even in result section, the reasons of experimental result were partly discussed which seems like very similar function of discussion section. And also, to discuss the result, in discussion section summary of results was suggested once again. This makes the manuscript very long and not clear. Hope authors reconsider the structure of manuscript and find out effective way to deliver what this paper really would like to say. One of ways is to combine two sections. I would like to suggest good example with good structure, Piao et al.,2018. Please consider the structure of the manuscript.
   We re-structured and shortened the manuscript following the suggestions of both reviewers. Specifically, we merged the Result and Discussion section following the advice of reviewer 2.

II.       Specific

1    Title can be reconsidered. Please see answer to I.4 above.

2    L5: Though the authors have done an experiment on global scale, the word "to simulate **local** atmospheric δ13C(CO2)" can make readers misunderstand that this result can have a bias from scale differences. It would be good to mention just "atmospheric δ13C(CO2)" minus local or put more appropriate word in place of "local" (for example global background stations?)
We removed the word "local" and added "at globally distributed sites to read: "..to simulate atmospheric $\delta^{13}C(CO_2)$ at globally distributed monitoring sites."

3    L24: It might be wonder δ13C is same to δ13Ca in Line 20. If it is same, please revise and unify all terms in the manuscript. e.g. L27 as well.
The term "$\delta^{13}C$ data" on L24 refers to isotopic data in general and not just for the atmospheric reservoir. Text on L27 and at other places modified as requested.

4    L25: Recommend that authors can suggest definitions of δ13C before mentioning 13C and 12C, especially if δ13C was used differently from δ13Ca (for example: 13C/12C, normally expressed relative to a standard as δ13C(CO2) in units of per mille (‰)).
The definition of $\delta^{13}C$ is now given on L20.

5    L78: 19 sites represent of global levels? To avoid the question of error of this experiment, the reason to choose 19 sites can be discussed somewhere in section 2.3 and here explicitly mention as 19 global sites.
Text modified to read: "..at 19 globally distributed sites"

6    L82: This sentence is not suitable for introduction. Introduction is not abstract. This is one of results in this manuscript. Hope it can be moved to another section or removed. *We demonstrate for the first time that the observations at the globally distributed sites show no significant trends in the seasonal cycle amplitude of δ13Ca, consistent with our model chain, but surprising in view of the large trend in the seasonal amplitude of CO2.*
Sentence removed as requested.

7    Section 2.1. There are too many abbreviations without full name. L92 (EMIC Bern3D-LPX), L93(Bern3D), L94(LPX), L98(DIC), L18 (LUH2 and NMIP), L120 (NCEP/NCAR), L121 (CRU-TS4.05).
L92: EMIC is defined on L79 as Earth System Model of Intermediate Complexity. For clarity, the subsection title is modified to read "Bern3D-LPX Earth System Model of Intermediate Complexity" and L92 modified to read. "… are simulated with the Bern3D-LPX Earth System Model of Intermediate Complexity."
L92/L93: Bern3D is a model name and not an abbreviation.
L94: LPX is now defined as Land surface Processes and eXchanges (LPX) model.
L98: "DIC" is replaced by "dissolved inorganic carbon".
L118: "LUH2" is replaced by "the Land-Use Harmonization 2 dataset"; "NMIP" is replaced by "$N_2O$ Model Intercomparison Project
L120: Sentence modified to read: "The monthly wind stress climatology from the NCEP/NCAR Reanalysis produced by the National Centers for Environmental Prediction (NCEP) and the National Center for Atmospheric Research (NCAR) …"
L121: Sentence modified to read: "Climatic Research Unit (CRU) Time-Series (TS) version 4.05 of high-resolution gridded data of month-by-month variation in climate (CRU TS4.05) \citep{Harris2020} are used for the land model."

8    L123: agents to species. Replaced.

9     L123: the specific definition of Econtrol.
      Text modified to read "A control simulation, termed $E_{control}$, .."

10    L125: Atmospheric CO2 to Ca, δ13C to δ13Ca. Done.

11    L127: what does TM3 stand for?
      TM3 is a model name. We use now the wording given in the publication by Heimann and Körner, 2003: "the global atmospheric tracer model TM3, a three-dimensional transport model".

12    L129 and L130: Just write Ca and δ13Ca because they were defined already in previous section. Done.

13    L135: 13C or δ13Ca ? Those confusions occurred all manuscript.
      The term $^{13}C$ is correct as we discuss the net atmosphere-to-surface flux and the signature of this flux $\delta^{13}C_{as,net}$, and not the atmospheric signature $\delta^{13}C_a$.

14    L140: Ca and δ13Ca? If not, I think those are also redefined as similar format. For example, δ13Co (ocean) or δ13Co (observation) etc. This is similar to the L142 and 143.
      The text is correct but we modified the wording to improve clarity. Bern3D-LPX simulates the flow of $CO_2$ and $^{13}CO_2$ from the ocean to the atmosphere and from the atmosphere to the ocean and similar for the land. We are concerned with the resulting net flux from the atmosphere to the surface as in Eq. (1) and not with atmospheric $CO_2$ ($C_a$) nor the isotopic signature of the atmospheric $CO_2$ ($\delta^{13}C_a$). The sentence on L139/140 is modified for clarification to read: "Bern3D-LPX simulates two-way exchange of $CO_2$ and $^{13}CO_2$ from the atmosphere to the ocean and land surface and from the ocean and land surface to the atmosphere."

15    L152 to 153: The authors should mention the URL with last access date from Cooperative Global Atmospheric Data Integration Project and Scripps CO2 program. I highly recommend adding that information. This is very important part.
      The URLs and last access dates are provided on L530 in the section "Code and data availability".

16    Section 2.3: Authors should explain the reason why 19 sites were selected for the experiment and their measurement uncertainty. Also NOAA data and Scripps data have different scale and there is a bias between two data derived from two scales (Lueker et al., 2020)
      We used the sites for which both data and transport matrices are available.
      Sentence revised to read: "Background CO2 from 19 monitoring sites, for which transport matrices are available, is used .."
      We note now the different scale: "The Scripps and GLOBALVIEW-CO2C13 data are on a slightly different scale \citep{Lueker20}; this does not affect our analysis of seasonal anomalies.

17    L160: If authors did not use KER and NZD data, it would be good to not discuss here to avoid confusions.
      Sentence removed as requested.

18    L162: Hope you can keep the same term all over the manuscript, for Ca and δ13Ca.
      We checked the manuscript and use the same terms where appropriate.

19    L166: 13C to δ13Ca?
      $^{13}C$ is correct here as equation (3) displays the budget for the atmospheric inventory of $^{13}C$.

20    L191: Background CO2 mixing ratio is different from observed CO2 mixing ratio. This is important part because just observation CO2 include local signals but background CO2 is selected representative values from all observation data. Authors should discuss this in data section 2.3 as indicating which data were used for the experiment. Also please do not use CO2 mixing ratio in place of Ca. If authors defined Ca, please keep the term. Also the unit of data that are used for this paper are not mixing ratio, that is mole fraction (Green book, 2007, Note. Official name is not green book but normally use as green book.)
      Sentence at beginning of section 2.3 Site data revised to read: "Background $CO_2$ from 19

monitoring sites …"
We now use the term mole fraction instead of mixing ratio in the text. The difference between mixing ratio and mole fraction is very small for trace gases like $CO_2$
We now use the term $C_a$ for atmospheric $CO_2$ where appropriate throughout the text.

21  L206: The model…, there are many models in the method section. It would be good to indicate explicitly what model is. For L212 The land biosphere model and L216 The ocean model should be explained for what kind of models were used, as well. Also, it would be good to match all experiment results with 2. Method section.
L206/212/216: Model names (Bern3D-LPX, LPX, Bern3D) added.
L206: text modified to read "..simulates in the standard setup ($E_{standard}$) .." to indicated model setup us requested.

22  L244: Again, δ13C and CO2 are differed from Ca and δ13Ca?

Text is correct as is as it refers to atmosphere-surface fluxes and not to the atmosphere.

23  Table 1: It would be helpful to display the stations according to the latitude. Maybe swap Mahe Island and Acension Island?
Order changed in Tab. 1 and Tab. 2.

24  L227: Does "from 1982 to 2012" mean Estandard? Keep the term in whole manuscript.
Sentence adjusted to read: " Figure 2 compares the mean seasonal cycles of Ca and δ13Ca from Estandard with measurements from 1982 (Alert: 1985) to 2012 at three sites .."

25  L234: Does "Standard simulation" mean Estandard?
Term replaced by $E_{standard}$

26  L254: South Pole, Palmer, and Halley, L255: Figs.2, S1, and S2.
L254/L255: "and" added as requested.

27  Figure 3. It would be good to add latitude information next to the name of stations. Also next to Panel (a), 'Data from Scripps…' can be removed.
We provide now the full sites names and link them with the abbreviations used in panel a) as well as the latitude of each site in the captions.  The title of panel (a) was removed.

28  L284: the unit 'permil/century' is difficult to understand through Fig.3.
Changed labels to "permil"

29  Table 2: Can they be revised?; Observation data CO2 to observation Ca, GlOBALVIEW-CO2 to CO2, SCRIPPS to Scripps.  Done.

30  L291: SA(δ13Ca) Done.

31  L294: This can be moved to Method section.
*The Scripps data, including seasonality, are provided as (i) monthly samples, (ii) a fit to these monthly samples, and (iii) the monthly samples but missing values replaced with fitted values. We also used the original, non-gap-filled data and years with at least 9, 10, or 11 monthly values per year in the regression*
Done.

32  L311: Why do authors analyze model and observation slope? Please add the purpose.
We added:".. to probe model-observation agreement".

33  L326: SA means SA(Ca) or SA(δ13Ca)? Or both of SA?
"*SA*" replaced by "*SA($C_a$)* and *SA($\delta^{13}C_a$)*".

34  L335: Authors mentioned only the diverse range of SA, but the values seem like very significant. The explanations are focused on they are reliable data rather than the meaning of values. Was this discussed somewhere in the manuscript?
Yes, the implications are discussed in section 5.1 L406 of the original manuscript. The text "The seasonal amplitude of $CO_2$ (*SA($C_a$)*) is observed to grow over time depending on location

(Bacastow et al., 1985; Barlow et al., 2015; Piao et al., 2018) and driven by changes in the seasonality of net land carbon uptake (Graven et al., 2013; Forkel et al., 2016)." has now been moved to the introduction. As the manuscript is already long and the focus is on $^{13}$C, we do not further discuss trends in $SA(C_a)$.

35  L336: Does 'industrial period' mean Estandard?
"$E_{standard}$ minus $E_{control}$" added.

36  L350: Does 'pre-industrial to the reference period' mean that 'Econtrol to Estandard'?
No. The text refers to observed increase in atmospheric $CO_2$.

37  L363 to L374: Can we move whole part to Method section? Or combine to Appendix A?
We prefer to keep Eq. 9 in this subsection and next to the text on the decomposition and Fig. 4. This allows the reader to link the results presented in this subsection and in Fig. 4 with Eq. 9.

38  L406 to L441: Those are explained already in Section 4 and more similar to summary rather than discussion section. Only differences are adding references more. It would be good to make it simple and clear as suggested general review.
Done. Text has been shortened and integrated into previous sections.

39  L409: What is the number of 'relatively small uncertainties'? and for 'no clear trend in the standard case' in L423.
Text on L409 and L423 deleted during the revision.

40  L423, L425: If 'standard case' and 'preindustrial control' mean that Econtrol and Estandard, please keep the same therm.
The text has been deleted during the revision.

41  L445: NPP is different from the NPP in section 4.3.2? If same, why do authors invite another term, εNPP, here?
Text shortened and clarified to read: "Following Farquhar (1989) and Cernusak et al. (2013) $\varepsilon_{NPP}$ is .."

42  Section 5.2: I have quite similar opinion to section 5.1. The manuscript was mixed with result (discussed before) and seems like more conclusion section? It is very vague what authors really would like to say.
The text in section 5.2 has been shortened in incorporated into earlier subsections.

Reference
Piao et al.(2017) https://doi.org/10.1111/gcb.13909
Lueker et al., (2020) https://escholarship.org/uc/item/4n93p288
Greenbook(2007) https://iupac.org/wp-content/uploads/2019/05/IUPAC-GB3-2012-2ndPrinting-PDFsearchable.pdf

**RC2: Gerbrand Koren, 28 Aug 2024**

I have enjoyed reading the manuscript entitled "No increase is detected and modeled for the seasonal cycle amplitude of δ13C of atmospheric carbon dioxide" by Joos and co-authors.

Their study looks into the seasonal cycle at a number of stations across the globe for CO2 and δ13C(CO2). The authors find no trend in the seasonal cycles of atmospheric observations and simulations from their modeling framework. The model also allows them to assess (iso)fluxes and their spatiotemporal patterns.

Overall I feel that this is a thorough study and I expect that it will be a valuable resource for the community. My main concerns are related to missing connections with some other key δ13C studies and the length and structure of the manuscript. I recommend publishing the manuscript after addressing these comments.

We thank Koren Gebrand for his careful review and his highly useful advice that helped us to improve the presentation of our findings.

MAIN COMMENTS

(1) CONNECTION WITH KEELING ET AL. (2017)

The key paper by Keeling et al. (2017) is not discussed. I believe that it is a relevant reference at various places in this manuscript, in particular the Introduction and Sect. 5.1. A few specific examples (not-exhaustive) are provided below:

-L9: "no long-term temporal changes in the isotopic fractionation by C3 plants." This seems to contradict with Keeling et al. (2017).
We now discuss the findings of Keeling et al. in the main text. These authors inferred changes in discrimination of global mean net primary production, whereas we discuss on L9 northern extratropical regions. We clarified the text to read: " ..with no long-term temporal changes in the isotopic fractionation in these ecosystems dominated by C$_3$ plants."

-L48-50: "It remains to be assessed whether a scenario with small long-term changes in fractionation of C3 plants is compatible with atmospheric δ13Ca observations representing carbon fluxes over large regions." How does this relate to Keeling et al. (2017), who conclude that there is a long-term change in discrimination?
We deleted the sentence on L48-50 and the related sentence starting on L47 ("Upscaling of results from site studies to large scales is challenging") and added: "Keeling et al. (2017), analyzing decadal-scale change in seasonally detrended δ$^{13}$C$_a$ and the annual atmospheric budgets of carbon and $^{13}$C, find a decrease in isotopic fractionation of global mean net primary production; the change is attributed to changes in fractionation associated with mesophyll conductance and photorespiration of C$_3$ plants and intrinsic water use efficiency is inferred to grow proportionally with C$_a$. "

We added the word "seasonality" to the third bullet point in the introduction to read: "Is a model scenario with intrinsic water use efficiency growing proportional with C$_a$ consistent with δ$^{13}$C$_a$ seasonality data?"

Further, we now briefly discuss the finding of Keeling et al. at the end of the original section 4.3.2.

Keeling et al. (2017) analyzed the atmospheric budgets of carbon and $^{13}$C over the period 1975 to 2005 and inferred a change in discrimination for the global mean net primary productivity (NPP) of 0.66 ±0.34 permil. These authors applied a three-box biosphere model coupled with a one-box atmosphere

and a box-diffusion ocean model. The model does not distinguish between $C_3$ and $C_4$ photosynthesis and features time-invariant land carbon overturning rates and a net primary production (NPP) of around 50 GtC $yr^{-1}$, increasing linearly with $CO_2$.

A recent analysis using bomb-radiocarbon data as constraints suggests a much larger NPP of presently more than 80 GtC $yr^{-1}$. The NPP applied in the 3-box model of Keeling is also at the lower end of the range from current land models (46-76 GtC/yr). A higher NPP implies larger disequilibrium fluxes, potentially offsetting the contribution from the postulated trend in discrimination.

Substantial uncertainties in the atmospheric $^{13}C$ budget are linked to uncertainties in fossil fuel emissions and its signature. The fossil fuel flux increased by 100 GtC permil from 129 to 228 GtC permil from 1975 to 2005. It remains difficult to quantify uncertainties in the temporal evolution of this flux, adding uncertainty to the estimate of Keeling et al.

Land use reconstructions show a shift in the distribution of C3 versus C4 plants. LPX-Bern simulations with prescribed land use and crop distribution suggest a change in fractionation of global mean NPP of about 1.5 permil over the industrial period associated with changes in $C_3$ versus $C_4$ plant distribution, larger than assumed by Keeling et al. in a sensitivity analysis.

We added the following text in section 4.3.2: "Keeling et al. (2017) analyzed the atmospheric budgets of carbon and $^{13}C$, using seasonally detrended data, a three-box land model with time-invariant overturning timescales, globally uniform isotopic fractionation, and neglecting changes in $C_3/C_4$ distribution in their standard setup. They found global mean $\varepsilon_{NPP}$ to decrease by 0.66±0.34‰ from 1975 to 2005 and attributed this change to changes in fractionation associated with mesophyll conductance and photorespiration of $C_3$ plants. It appears challenging to detect and attribute changes in the fractionation of global mean NPP with a box model, given uncertainties in NPP (Graven et al., 2024) and changes in C3 versus C4 plant distribution."

We also expanded the text on variations of $\varepsilon_{NPP}$ in this section by adding:" …while $\varepsilon_{NPP}$ remains time-invariant in the $C_3$-dominated ecosystems north of 45°N (Fig. 4c)."

-L451: "An absent temporal trend in ci/ca translates into an absent trend in εNPP, and vice versa (Eq. 10)." What if the other terms of Eq. 1 in Keeling et al. (2017) are also considered?

We now discuss the simplification with reference to the review on isotopic fractionation by Cernusak et al, 2013 and the work by Farquhar and Cernusak,2012. This additional discussion does not alter our conclusions.

The implementation of fractionation in LPX-Bern is clarified in the method section to explicitly mention the fractionation during dissolution and water transfer and by photorespiration, two factors discussed by Keeling et al.: "The scheme does not explicitly consider fractionation by boundary layer transport and ternary effects associated with the interaction of $CO_2$, water, and air (Cernusak and Farquhar 2012) and fractionation by "dark" day respiration is set to zero, while fractionation by the following terms is explicitly considered: stomatal conductance (with a scaling factor of 4.4 permil), dissolution and liquid transport (1.8 permil), carboxylation (27.5 permil), and photorespiration (8 permil and the $CO_2$ compensation point that would occur in the absence of dark respiration, $\Gamma^*$, is increasing with temperature)."

We added the word approximately to the sentence before Eq. 10: "Following Farquhar (1989), the fractionation for $C_3$ photosynthesis and NPP ($\varepsilon_{NPP}$) is approximately proportional to $c_i/c_a$:"

We added the following text to the original section 5.1/new section 4.4:" Equation (10) is an approximation (Farquhar et al.,1982, Llyod&Farquhar, 1994, Farquhar&Cernusak, 2012, Cernusak et

al. 2013) considered to be sufficient for many applications by Cernusak et al. 2013 and applied in the publications cited in the previous two paragraphs. However, there are four contributions only implicitly considered by choosing parameter $b$ in Eq. 10 and these may contribute small temporal trends to $\varepsilon_{NPP}$. In turn, inferred $c_i/c_a$ would also have a temporal trend for a constant $\varepsilon_{NPP}$. We estimate the trend contribution of these additional terms to be of small magnitude (<1 permil) in comparison to the 3 to 3.8 permil difference estimated for our two scenarios (see Appendix B) for details)."

We added the following text in the Appendix: "Appendix B: Uncertainties in the relationship between $\varepsilon_{NPP}$ and $c_i/c_a$"

In section 4.4, we applied a simplified expression for fractionation of $C_3$ plants during photosynthesis ($\varepsilon_{NPP}$) and used this expression to translate trends in $\varepsilon_{NPP}$ to trends in $c_i/c_a$ and in $iWUE$. The potential contributions to trends in $\varepsilon_{NPP}$ from neglected ternary effects, "dark" day respiration, and transport through the mesophyll and photorespiration are discussed in this appendix.

Isotopic fraction for $C_3$ photosynthesis is framed as a multi-step process considering the transport of $CO_2$ and the underlying gradients in $CO_2$ mole fractions, from the ambient air (mole fraction: $c_a$) to the leaf surface ($c_s$) in the intercellular air spaces ($c_i$) and the sites of carboxylation ($c_c$) plus the fractionation during carboxylation, "dark" day respiration, $R_d$, and photorespiration (Cernusak 2013). The transport of $CO_2$ equals the consumption of $CO_2$ by assimilation, $A$: $A = g (c_a - c_i) = g_m (c_i - c_c)$, with $g$ being the conductance of the stomatal pores and the boundary layer and $g_m$ the mesophyll conductance. The relationship can be rewritten as $A/(g c_a) = (1 - c_i/c_a) = g_m/g (c_i/c_a - c_c/c_a)$. If $A$ is increasing in proportion to $c_a$ and $g$ and $g_m$ assumed constant, then it follows that also $c_i/c_a$ and $c_c/c_a$ are constant. In turn, the fractionation associated with boundary layer and stomatal conductance ($-a (1 - c_i/c_a)$; $a = 4.4‰$), mesophyll conductance ($-a_m (c_i/c_a - c_c/c_a)$; $a_m = 1.8‰$), and carboxylation ($-b \times c_c$) remain constant. The overall influence of mesophyll transport on $\varepsilon_{NPP}$ can also be written as $(b - a_m)/g_m \times A/c_a$ (Keeling et al., 2017).

Keeling et al. (2017) assumed that $A/c_a$ decreases over time, with $A$ increasing by 45% for a doubling of $CO_2$, and that therefore fractionation by the mesophyll contribution would change by -0.006‰ ppm$^{-1}$, i.e., a change in $\varepsilon_{NPP}$ of 0.47‰ for the $CO_2$ increase of 78 ppm from 1980 to 2022. On the other hand, Campbell et al. (2017) observationally constrained the growth in gross primary production over the 20th century to be 31±5%, larger than the increase in $c_a$ of 25%. Accordingly, $A/c_a$ increases and the mesophyll trend contribution is positive. With the central parameters values of Keeling et al. ($A$=9 µmol m$^{-2}$ s$^{-1}$, $g_m$=0.2 mol m$^{-2}$ s$^{-1}$, $CO_2$=355 ppm) the contribution is +0.002‰ ppm$^{-1}$. Keeling et al. also estimated changes in fractionation associated with photorespiration ($-f \times \Gamma^*/c_a$; $f = 12‰$) to -0.004‰ ppm$^{-1}$ assuming a constant $CO_2$ compensation point, $\Gamma^*$. The real sensitivity must be smaller as $\Gamma^*$ increases with temperature and because Keeling et al. applied an estimate for the $CO_2$ compensation point in the presence of $R_d$ (43 ppm) instead of the absence of $R_d$ ($\Gamma^*$=31 ppm). Further, fractionation during day respiration is $-e \times c_c/c_a \times R_d/V_c$ (Cernusak et al., 2013), roughly about 0 to -0.3‰ for $e$ in the range of 0 to 5‰; we apply a Rubisco carboxylation rates, $V_c$, of 11 µmol m$^{-2}$ s$^{-1}$ derived from the value of $A$=9 µmol m$^{-2}$ s$^{-1}$ by Keeling et al., $R_d$=1 µmol m$^{-2}$ s$^{-1}$, and $c_c/c_a = 0.6$). Finally, ternary effects of about -0.7‰ (0.024 × b) increase with water vapor deficit (Farquhar and Cernusak, 2012). Given the small amplitudes of these two contributions, their temporal trends are likely also small over recent decades."

We further corrected the sign of equation 10.

(2) LEARNING FROM dC13(CO2)

The authors make various statements about the potential of the seasonal variation of δ13C to constrain biosphere models. I appreciate the big picture that the authors create, but the statements would be more convincing if authors would also provide some more direction on how to achieve this. Also, some reflection on studies that already attempt this would strengthen the arguments.

-L14-16: "We propose to apply seasonally-resolved δ13C(CO2) observations as a novel constraint for land biosphere models and underlying processes for improved projections of the anthropogenic carbon sink." and L525,526: "We recommend to apply seasonally-resolved δ13Ca observations as a novel constraint for land biosphere models used to simulate the terrestrial sink of anthropogenic carbon and land use emissions." How should this constraint be used?

L14/16: We replace "a novel constraint" with "an additional constraint"

L525: We modified the sentence to read: "We recommend applying seasonally-resolved $\delta^{13}Ca$ observations as a constraint for land biosphere models used to simulate the terrestrial sink of anthropogenic carbon and land use emissions, for example, by using perturbed parameter ensembles in Bayesian approaches (Lienert et al., 2018, Van der Velde et al., 2018)."

-L69,70 :"... but to our knowledge have not been used as a benchmark for model performance in combination with an atmospheric transport model and for analyzing trends in SA(δ13Ca) globally." I think the study by van der Velde et al. (2018) does this through data assimilation in their model framework. It would be good to reflect on that, and what more could be done. Finally, also Ballantyne et al. (2011) is not mentioned. Those authors reflect on seasonality of δ13C and use δ13C to learn about some leaf parameterisations.

We modified the text as follows:
L61: we replaced "lacking" with "scarce" and deleted "to our knowledge" to read: "Comparable studies, analyzing the temporal trends in SA( δ 13Ca) and the seasonal cycle of δ 13Ca are scarce."
L67: We added the following text:" Van der Velde et al. (2018) applied their Carbon Tracker Data Assimilation System for $CO_2$ and $^{13}CO_2$ by varying the net exchange fluxes of $CO_2$ and $^{13}CO_2$ in ocean and terrestrial biosphere models and propagating the fluxes through an atmospheric transport model to solve for weekly adjustments to fluxes and isotopic terrestrial discrimination minimizing differences between observed and estimated mole fractions. They identified a decrease in stomatal conductance on a continent-wide scale during a severe drought. Ballantyne et al., 2011 applied an analytical regression approach to analyze the differences in isotopic signatures between northern hemisphere site data versus free troposphere background data from Niwot Ridge to infer seasonal variations in the source signature of the net atmosphere-land biosphere flux and to evaluate models of stomatal conductance."
L 69/70: We deleted: "as a benchmark for model performance in combination with an atmospheric transport model and"

(3) LENGTH AND STRUCTURE

Overall, I found the manuscript quite lengthy especially Sects. 4 (Results) and 5 (Discussion). Based also on the titles of the subsections there appears to be some overlap in the scope of different subsections. I would recommend shortening and potentially integrating the Results and Discussion section such that in one of those subsections a certain aspect can be described more holistically, avoiding some (perceived) overlap.

We followed the advice and integrated the Results and Discussion sections and shortened the manuscript.

Also some paragraphs could be moved to other sections or the supplement, e.g.: L505-508: "On a technical note, transporting simulated 13C fluxes is not without challenges. The definition of the δ-notation can pose numerical difficulties when net 12C fluxes are close to zero. We find that transporting signature-weighted total carbon fluxes is the most reliable method for arriving at local δ13Ca. Similarly, seemingly small errors in the model representation of gross fluxes and mass balances, can become critical when considering net surface-to-atmosphere fluxes." This seems to be a valuable comment, but a bit strange to end the Discussion with this point. This can probably be integrated in a more natural way in the Methods section, e.g. somewhere around Eq. 1.
The paragraph is deleted for brevity. The two approaches are briefly discussed in section 2.2.

MINOR COMMENTS

L20: For completeness I recommend to include a definition of δ (this is e.g. needed for the derivations in section 3).
Done.

L181-182: "In this way, a positive (negative) flux causes a positive (negative) change in δ13Ca". Should "positive (negative)" in the latter part of the sentence be reversed to "negative (positive)", because of the minus in the equation?
Corrected.

Table 1 and 2: The labels "Standard" (and "Std") are a bit confusing when quickly looking at these tables. I recommend using "Simulated" (and "Sim"), or "Modeled" (and "Mod") to be consistent with the in-text equation in L311-312.

Done. We use "Model" and "Mod".

Fig. 4: Is "/yr" missing from the unit on the y-axes for panels a and b?
No. As shown in Eq. 7 and 8 and described in Appendix A, the seasonal amplitude of a flux is defined as the integrated flux over the growing season. We added a reference to Eq. 8 in the caption of Fig. 4

L557: Here you mention why you used NPP and not GPP. I think this should be stated much earlier, for both Eqs. 9 and 10.
Done. Sentence moved to the paragraph with Eq. 9.

SPECIFIC COMMENTS

L31: "earth system models", capitalize? Done.
L32: "C3" is usually written with subscript (throughout manuscript, similar for "C4") Done.
L58: "(e.g. Peylin et al. (2013))" > "(e.g. Peylin et al., 2013)" Done.
L95: "(about 9ox4.5o)", replace letter "o" with degree symbols, as in e.g. L100 (similar for L131,137) Done.
L95: "(about 9ox4.5o)", replace letter "x" with multiplication symbol ("×"), also for other lines Done.
L181: after "instead" insert "of" Done.
L186, Eq. 7: Move "dt" after "Fas,net(t)"? Done.
L188, Eq. 8: Move "dt" after "d13F*as,net(t)"? Done.
L200: "These seasonal fluxes will be presented in section 3.3." Section 3.3 does not exist
Reference corrected.
L280: Fig 3, title above panel a misses "("
title removed as suggested by reviewer 1.

L323,324: "represents well the (...) atmosphere", change to ""represents the (...) atmosphere well"", or alternatively "accurately represents the (...) atmosphere""
Done.
L389: "4;see" > "4; see" Done.
L498: "δ13C(CO2)" > "δ13Ca" Done.
L502 (2x): "δ13C(Ca)" > "δ13Ca"  Done.
L510: "δ13C(CO2)" > "δ13Ca" Done.
L514: "δ13C(CO2)" > "δ13Ca" Done.
L515: "δ13C(CO2)" > "δ13Ca" Done.
L519: "δ13C(CO2)" > "δ13Ca" Done.
L519: "tropic" > "tropical" Done.
L536: "(e.g., Mook (1986); Joos and Bruno (1998))" > "(e.g., Mook, 1986; Joos and Bruno, 1998)" Done.
L708: "$CO_2$" > "CO2" Done.
L731: "Keeling, C. D., B., B. R.," > "Keeling, C. D., Bacastow, R. B.," Done.
L733: "https://doi.org/10.1029/GM055p0277" > https://doi.org/10.1029/GM055p0165  Done.

REFERENCES

Ballantyne et al. (2011). Novel applications of carbon isotopes in atmospheric CO2: what can atmospheric measurements teach us about processes in the biosphere? Biogeosciences, 8, 3093–3106. https://doi.org/10.5194/bg-8-3093-2011

Keeling et al. (2017). Atmospheric evidence for a global secular increase in carbon isotopic discrimination of land photosynthesis. Proceedings of the National Academy of Sciences, 114(39), 10361–10366. https://doi.org/10.1073/pnas.1619240114

van der Velde et al. (2018). The CarbonTracker Data Assimilation System for CO2 and δ13C (CTDAS-C13 v1.0): retrieving information on land–atmosphere exchange processes. Geoscientific Model Development, 11, 283–304. https://doi.org/10.5194/gmd-11-283-2018

Thanks for the opportunity to review this work.

**Citation**: https://doi.org/10.5194/egusphere-2024-1972-RC2